# *ANKS1B* encoded AIDA-1 regulates social behaviors by controlling oligodendrocyte function

Chang Hoon Cho[1,8,9], Ilana Vasilisa Deyneko[1,9], Dylann Cordova-Martinez[1], Juan Vazquez[1], Anne S. Maguire ●[1], Jenny R. Diaz[1], Abigail U. Carbonell[1], Jaafar O. Tindi[1], Min-Hui Cui[2,3], Roman Fleysher[2,3], Sophie Molholm[1,4,5], Michael L. Lipton[1,2,3,5], Craig A. Branch[2,3], Louis Hodgson ●[6,7] & Bryen A. Jordan ●[1,5] ✉

Heterozygous deletions in the *ANKS1B* gene cause ANKS1B neurodevelopmental syndrome (ANDS), a rare genetic disease characterized by autism spectrum disorder (ASD), attention deficit/hyperactivity disorder, and speech and motor deficits. The *ANKS1B* gene encodes for AIDA-1, a protein that is enriched at neuronal synapses and regulates synaptic plasticity. Here we report an unexpected role for oligodendroglial deficits in ANDS pathophysiology. We show that *Anks1b*-deficient mouse models display deficits in oligodendrocyte maturation, myelination, and Rac1 function, and recapitulate white matter abnormalities observed in ANDS patients. Selective loss of *Anks1b* from the oligodendrocyte lineage, but not from neuronal populations, leads to deficits in social preference and sensory reactivity previously observed in a brain-wide *Anks1b* haploinsufficiency model. Furthermore, we find that clemastine, an antihistamine shown to increase oligodendrocyte precursor cell maturation and central nervous system myelination, rescues deficits in social preference in 7-month-old *Anks1b*-deficient mice. Our work shows that deficits in social behaviors present in ANDS may originate from abnormal Rac1 activity within oligodendrocytes.

Despite evidence for a large genetic contribution to neurodevelopmental disorders (NDDs), defining causal pathways is difficult because individual mutations typically exert only mild effects and the risk for disease increases with multiple mutations[1]. NDDs associated with copy number variations (CNVs) or other mutations predicted to have major effects on protein function[2], especially those associated with CNVs affecting single genes (monogenic), are critically important for research because they can establish a direct link between one gene and a set of cellular and behavioral phenotypes. We recently identified individuals around the world with heterozygous and monogenic deletions in the *ANKS1B* gene that cause haploinsufficiency[3], confirming a previously uncharacterized link between *ANKS1B* haploinsufficiency and a novel neurodevelopmental syndrome we term *ANKS1B* Neurodevelopmental Syndrome (ANDS)[3]. Patients with ANDS exhibit a

[1]Dominick P. Purpura Department of Neuroscience, Albert Einstein College of Medicine, Bronx, NY, USA. [2]Department of Radiology, Albert Einstein College of Medicine, Bronx, NY, USA. [3]Gruss Magnetic Resonance Research Center, Albert Einstein College of Medicine, Bronx, NY, USA. [4]Department of Pediatrics, Albert Einstein College of Medicine, Bronx, NY, USA. [5]Department of Psychiatry and Behavioral Sciences, Albert Einstein College of Medicine, Bronx, NY, USA. [6]Department of Molecular Pharmacology, Albert Einstein College of Medicine, Bronx, NY, USA. [7]Gruss-Lipper Biophotonics Center, Albert Einstein College of Medicine, Bronx, NY, USA. [8]Present address: Human Pathobiology and OMNI Reverse Translation, Genentech, Inc., San Francisco, CA, USA. [9]These authors contributed equally: Chang Hoon Cho, Ilana Vasilisa Deyneko. ✉e-mail: bryen.jordan@einsteinmed.edu

range of NDDs, including attention-deficit/hyperactivity disorder (ADHD), motor impairments, speech apraxia and autism spectrum disorder (ASD), the latter of which is present in >60% of patients. The *ANKS1B* gene encodes for the protein AIDA-1, which is highly expressed in excitatory neurons of the central nervous system (CNS) and is one of the most abundant proteins at neuronal synapses[4,5]. AIDA-1 is localized at postsynaptic densities (PSDs), where it binds to N-methyl-D-aspartate receptors (NMDARs) and the synaptic scaffolding protein PSD95[6]. Excitatory neuron-specific (*Camk2a*-Cre) *Anks1b* knockout mice show reduced synaptic expression of the NMDAR subunit GluN2B and impaired hippocampal NMDAR-dependent synaptic plasticity[7], and CNS-wide (*Nestin*-Cre) *Anks1b* heterozygous mice display behavioral deficits across domains relevant to clinical features of ANDS[3]. Based on these findings, we previously postulated that neuronal dysfunctions and synaptopathies are involved in the pathogenesis of ANDS[3].

Despite a historical shortage on research into the role of white matter in brain diseases, recent literature has described myelin deficits early in the pathological cascade of prevalent disorders such as Alzheimer disease, Parkinson disease, and schizophrenia. In the process of investigating myelination in normal and abnormal CNS physiology, recent studies on oligodendrocytes have triggered rapid shifts in our understanding of this cell lineage. Oligodendrocyte precursor cells (OPCs) represent 5–10% of the glial population in the CNS and are increasingly recognized as active participants in normal and abnormal brain functions[8–13]. Oligodendrocytes are primarily involved in myelination of CNS axons, which emerging research reveals is dynamically regulated by sensory, motor, and cognitive activity in both humans and rodents throughout the lifespan[14–18]. There is also growing appreciation for the role of OPCs in the pathogenesis of neurodegenerative disorders including Alzheimer disease[19,20] and Parkinson disease[21]. OPCs have been implicated in ASD because altered white matter integrity[22–26] and reduced axonal conduction velocity[27,28] have been extensively documented in a subset of patients with this disorder. A recent study showed that haploinsufficiency of *Chd8*, a gene robustly linked to ASD etiology, leads to oligodendrocyte dysfunction and autism-related behaviors[29]. Recent large-scale single-cell transcriptomic initiatives suggest that AIDA-1 is expressed in oligodendrocytes and OPCs[30–33], as well as in newborn OPCs linked to lesions in multiple sclerosis (MS)[34]. Here, we test the hypothesis that oligodendrocyte dysfunction contributes to ANDS etiology.

Consistent with this speculation, we previously reported that magnetic resonance images (MRIs) from ~40% of ANDS patients reveal thinning or other structural anomalies of the corpus callosum, which is highly myelinated[3]. Here we reveal additional profound white matter abnormalities in ANDS patients that are also modeled by mice with *Anks1b* haploinsufficiency. Furthermore, we show that selective *Anks1b* ablation from oligodendrocytes is sufficient to reproduce both myelination deficits as well as behavioral correlates of ASD found in a brain-wide (*Nestin*-Cre) *Anks1b* haploinsufficiency model[3]. Surprisingly, these behaviors were not recapitulated by selective ablation of *Anks1b* from neuronal populations that highly express this gene. Moreover, we found that social deficits were rescued by treatment with clemastine fumarate, an FDA approved antihistamine used for seasonal allergies that was recently and unexpectedly found to increase myelination in a high-throughput screen for novel MS therapeutics[35].

Our previous proteomic and functional analyses reveal that AIDA-1 may be linked to small Rho family small GTPases, particularly to Rac1 function[36]. Rho GTPases are crucial for cytoskeletal dynamics[37], migration[38], and signal transduction[39], and have been shown to regulate the cellular morphology, maturation, and migration of oligodendrocyte lineage cells[40–44]. Additionally, dysregulation of these GTPases is linked to neurodevelopmental (ASD, Rett syndrome), neurodegenerative (Alzheimer, Parkinson), and psychiatric disorders[45–47].

Rac1 in particular has garnered significant attention due to its role in various neurodevelopmental[48,49] and neurodegenerative diseases[50] as well as its critical role in supporting oligodendrocyte functions including OPC migration, differentiation, and the formation of myelin sheaths around axons[40,41,51,52]. Dysregulation of Rac1 can therefore lead to faulty myelination and contribute to demyelinating disorders like MS[53]. Using FRET-based G-protein sensors, we confirm that loss of AIDA-1 affects Rac1 activation in primary cultured oligodendrocytes isolated from the *Anks1b*-deficient mice. We also show that that in vitro activation of Rac1 (as well as treatment with clemastine) rescues maturation deficits observed in these cells. Overall, our results suggest that the social and sensory behavioral correlates of ASD present in mouse models of ANDS originate from oligodendrocyte dysfunction.

## Results

### *Anks1b* haploinsufficiency impairs structural integrity of white matter

Several patients with ANDS had undergone brain MRI during prior clinical evaluation[3]. Radiologist impressions frequently cited abnormalities in the corpus callosum, among other findings in white matter (Fig. 1a). To substantiate the clinical observations, two ANDS patients underwent detailed Diffusion Tensor Imaging (3 Tesla-DTI) at the Gruss Magnetic Resonance Research Center at Albert Einstein College of Medicine to quantitatively assess integrity of white matter microstructure using Fractional Anisotropy (FA) (Fig. 1b, c). Patient FA values were compared to those of 101 neurotypical individuals previously collected as part of the Einstein Lifespan Study[54] (Fig. 1b–d, Supplementary Fig. 1). In both patients, we found multiple brain regions displaying FA values significantly ($p < 0.05$) below and above the average of the control group (Fig. 1d). These results reveal profound alterations in white matter microstructure throughout the brain in individuals with ANDS.

We had previously shown that CNS-wide (*Nestin*-Cre) *Anks1b* heterozygous mice (Nestin-Het mice) display behavioral correlates of patient phenotypes including hyperactivity, abnormal approach-avoidance behaviors, sensory hyperreactivity, impaired sensorimotor gating and fine motor coordination, and reduced social preference[3]. We performed volumetric analysis on T2-weighted MRI data to detect gross anatomical abnormalities and carried out DTI (Fig. 1e). Testing was performed on Nestin-Het and wildtype (WT) littermates for in vivo analyses ($n = 5$ per genotype), followed by ex vivo imaging of fixed brains of these and an additional set of mice ($n = 10$ per genotype). Volumetric analyses of ex-vivo brains revealed no statistically significant differences in overall brain volume between genotypes (Fig. 1f). However, Nestin-Het mice had a significant decrease in the volume of the corpus callosum (Fig. 1g), as well as in other highly myelinated brain structures including the optic tract (Supplementary Data 1). Moreover, in vivo DTI analyses revealed a significant decrease in FA of callosal (Fig. 1h) and other tracts (Supplementary Data 1), suggesting deficient white matter integrity and deficits in myelinated axons. These results show that mice with *Anks1b* haploinsufficiency model the white matter abnormalities found in patients.

### *Anks1b* haploinsufficiency results in impaired myelination and reduced oligodendrocyte abundance

To explore the white matter abnormalities observed in Nestin-Het mice, we performed histological and immunocytochemical analyses. Nissl staining confirmed that Nestin-Het mice have a thinner corpus callosum throughout the rostrocaudal axis (Fig. 2a), as well as larger lateral ventricles (LV) compared to WT littermates, although their dimensions were highly variable among animals (Supplementary Fig. 2a). Comprehensive analyses of other structures also revealed significant changes in the CA1/2 region of the hippocampus as well as in cerebellar lobule VI (Supplementary Fig. 2a, b), an area increasingly correlated with ASDs[55]. Since the decrease in callosal volume suggested impaired myelination, we stained brain slices with Luxol Fast

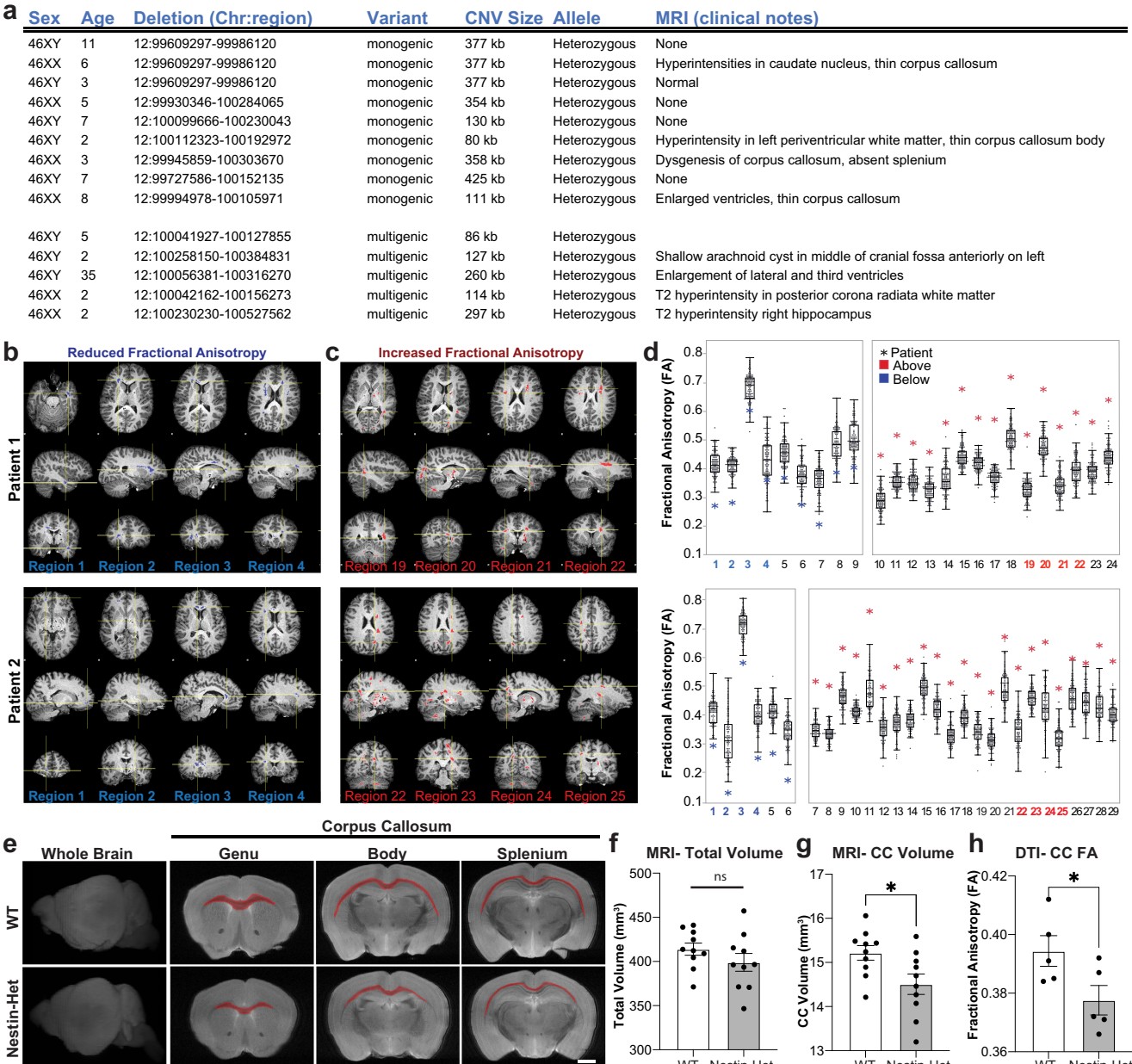

**a**

| Sex | Age | Deletion (Chr:region) | Variant | CNV Size | Allele | MRI (clinical notes) |
|---|---|---|---|---|---|---|
| 46XY | 11 | 12:99609297-99986120 | monogenic | 377 kb | Heterozygous | None |
| 46XX | 6 | 12:99609297-99986120 | monogenic | 377 kb | Heterozygous | Hyperintensities in caudate nucleus, thin corpus callosum |
| 46XX | 3 | 12:99609297-99986120 | monogenic | 377 kb | Heterozygous | Normal |
| 46XX | 5 | 12:99930346-100284065 | monogenic | 354 kb | Heterozygous | None |
| 46XY | 7 | 12:100099666-100230043 | monogenic | 130 kb | Heterozygous | None |
| 46XY | 2 | 12:100112323-100192972 | monogenic | 80 kb | Heterozygous | Hyperintensity in left periventricular white matter, thin corpus callosum body |
| 46XX | 3 | 12:99945859-100303670 | monogenic | 358 kb | Heterozygous | Dysgenesis of corpus callosum, absent splenium |
| 46XY | 7 | 12:99727586-100152135 | monogenic | 425 kb | Heterozygous | None |
| 46XX | 8 | 12:99994978-100105971 | monogenic | 111 kb | Heterozygous | Enlarged ventricles, thin corpus callosum |
| 46XY | 5 | 12:100041927-100127855 | multigenic | 86 kb | Heterozygous | |
| 46XY | 2 | 12:100258150-100384831 | multigenic | 127 kb | Heterozygous | Shallow arachnoid cyst in middle of cranial fossa anteriorly on left |
| 46XY | 35 | 12:100056381-100316270 | multigenic | 260 kb | Heterozygous | Enlargement of lateral and third ventricles |
| 46XX | 2 | 12:100042162-100156273 | multigenic | 114 kb | Heterozygous | T2 hyperintensity in posterior corona radiata white matter |
| 46XX | 2 | 12:100230230-100527562 | multigenic | 297 kb | Heterozygous | T2 hyperintensity right hippocampus |

**Fig. 1 | *Anks1b* haploinsufficiency leads to structural abnormalities in corpus callosum. a** Table showing patient genomic abnormalities and findings in clinical MRIs. MRIs from two patients showing sample brain regions of (**b**) reduced Fractional Anisotropy (FA) and (**c**) increased FA compared to controls. **d** Quantitation of FA in diverse brain areas showing patient values (asterisk) compared to the same regions in neurotypical controls (data points and box plots). Values shown = $p <$ 0.05 as determined by voxel-wise comparison using EZ-Map[141]. $N = 2$ affected individuals and 101 neurotypical controls. The range on box plots represents quartiles, center and whiskers delineate minima and maxima with outliers (3 standard deviations from the mean) plotted. Notice that most patient FA values lay beyond outliers. **e–g** T2-weighted MRIs of ex vivo brain tissue from 10 female *Anks1b* Nestin-Het and 10 female WT controls (age 3 months) reveal (**f**) no changes in overall brain volume, but (**g**) a significantly smaller corpus callosum. ($p = 0.02277$). Scale bar = 1 mm (**h**) In vivo diffusion tensor imaging (DTI) reveals decreased fractional anisotropy in corpus callosum ($p = 0.0493$), suggesting impaired white matter integrity. Nestin-Het ($n = 5$), WT ($n = 5$) for in vivo mouse experiments. **f–h** Error bars reflect mean ± s.e.m. T-tests, two-sided; *$p < 0.05$. Source data for **d**, **f–h** are provided as a Source Data file.

Blue (LFB), a copper phthalocyanine dye that binds to lipoproteins of the myelin sheath. Both male and female Nestin-Het mice had decreased LFB staining intensity compared to WT littermates in the corpus callosum (Fig. 2b) as well as in the cerebellum (Supplementary Fig. 2c). To directly measure myelination, we analyzed transmission electron microscopy (TEM) images of the corpus callosum in the sagittal plane. Quantification revealed a significant increase in g-ratio (signifying decreased myelination) in Nestin-Het mice in the most common axon diameters (0.3–1.7 μm) (Fig. 2c, d). A decrease in immunostaining for myelin basic protein (MBP) in the corpus callosum

(Fig. 2e) and cerebral cortex (Supplementary Fig. 2d) corroborated our TEM findings. We also observed fewer Olig2-positive oligodendrocytes in the corpus callosum (Fig. 2f). These results show that *Anks1b* haploinsufficiency reduces myelin sheath integrity and MBP expression, possibly by reducing the numbers of oligodendrocytes.

**Anks1b haploinsufficiency impairs oligodendrocyte maturation and migration**

To explore the mechanisms underlying these findings, we tracked the birth and maturation of oligodendrocyte lineage cells in Nestin-

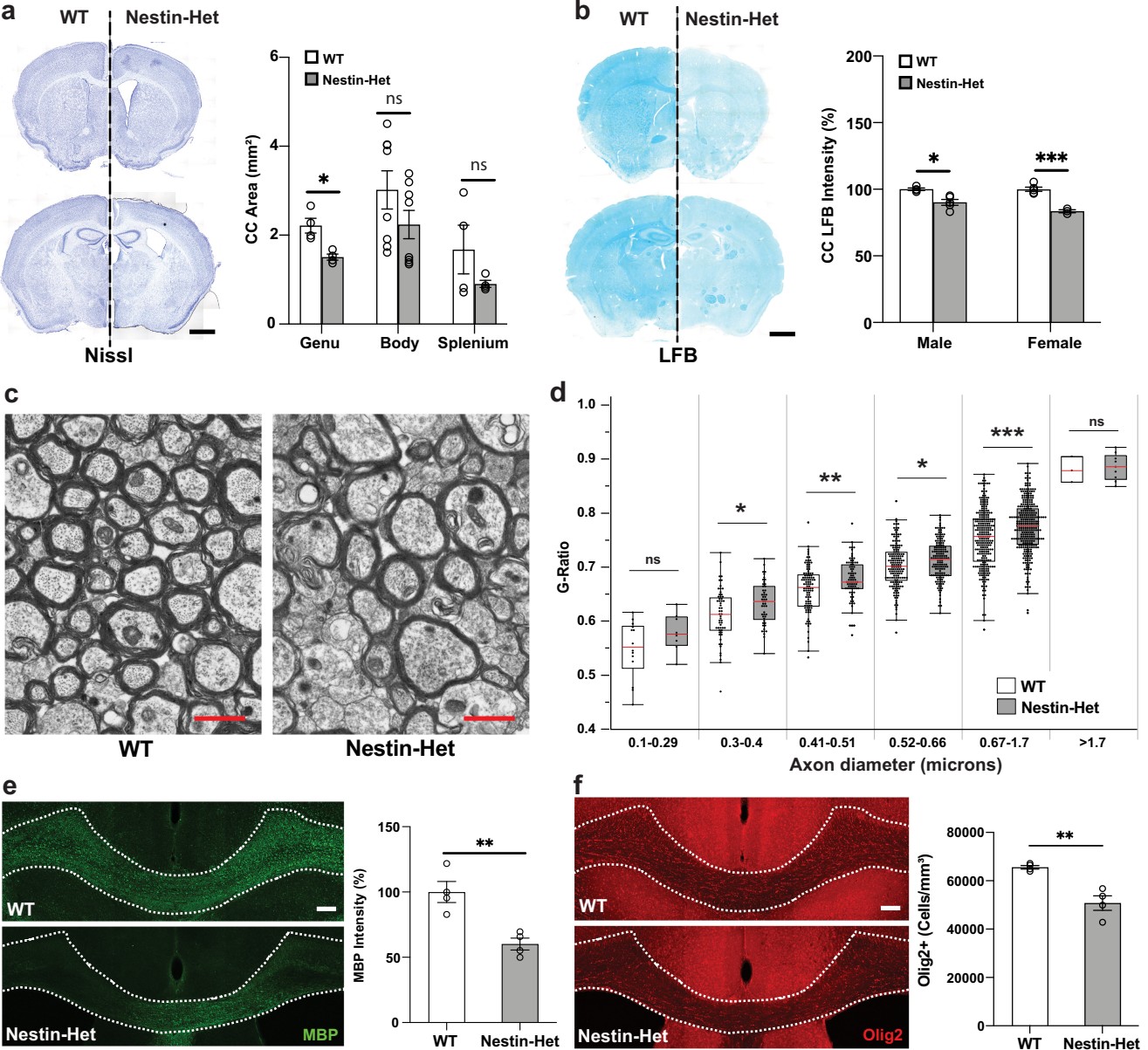

**Fig. 2 | Abnormal myelin sheath integrity and fewer oligodendrocytes in an *Anks1b* haploinsufficiency mouse model. a** Nissl staining of *Anks1b* Nestin-Het ($n = 6$) and WT ($n = 8$) mice shows decreased area of the corpus callosum. Images are representative of 18 sections from each mouse with acute slices from both genotypes placed side by side; scale bar = 1 mm; ($p = 0.007463$). **b** LFB (Luxol fast blue) staining for myelin sheath components reveals decreased density in corpus callosum of male and female *Anks1b* Nestin-Het mice compared to controls. Images are representative of 5 sections from each mouse; scale bar = 1 mm; (*$p = 0.006713$, ***$p = 0.000357$). **c** Transmission Electron Microscopy (TEM) images of callosal cross-sections from WT and Nestin-Het mice showing myelinated axons; scale bar = 1 μm. **d** Quantitation of g-ratio (the ratio of the inner-to-outer diameter of a myelinated axon) binned across the indicated axonal diameters. Results are from 12 animals ($n = 6$ WT, $n = 6$ Nestin-Het). Box plots show the 25th–75th quantiles (box), mean (red line), and whiskers delineate 3 standard deviations from the mean. **e** Immunostaining for myelin basic protein (MBP) reveals decreased myelination in callosal regions (dotted lines) of *Anks1b* Nestin-Het mice. Scale bar = 100 μm. ($p = 0.00523$). **f** Immunostaining for Olig2 reveals fewer oligodendrocytes per unit area. Scale bar = 100 μm. ($p = 0.00299$). For **e**, **f**: $n = 8$ *Anks1b* Nestin-Het, $n = 8$ WT, 4 sections from each mouse. **a**, **b**, **e**, **f** Error bars reflect mean ± s.e.m. **a**–**f** T-tests, two-sided; *, **, ***$p < 0.05, p < 0.005, p < 0.0005$. Source data for **a**, **b**, **d**, **e**, and **f** are provided as a Source Data file.

Het and WT mice. We injected bioconjugatable EdU intraperitoneally (IP) into pregnant mice at gestational age E18, and into mice 2, 6, and 12 months of age and tracked nucleotide incorporation over 4 weeks using markers of oligodendrocyte maturation to identify oligonucleotide lineage cells[11] (Fig. 3a). The cell surface tyrosine receptor kinase PDGFRa (platelet-derived growth factor receptor A) was used as a marker of OPCs and immature oligodendrocytes, and CC1 antibodies (targeting the RNA binding protein Quaking7[56]) were used to mark mature oligodendrocytes. Mice were sacrificed 4 weeks after EdU injections, fixed brain tissue was cut in coronal slices, and EdU-

positive newborn cells were labeled with reactive fluorophores. EdU-labeled mice at 3 weeks of age were obtained by injecting pregnant female mice at E18. Quantitation of labeled cells revealed significantly fewer newborn cells overall in the corpus callosum of Nestin-Het mice (Supplementary Fig. 3a), and significantly fewer mature oligodendrocytes (CC1-positive) derived from newborn cells throughout all time points tested (Fig. 3b). We also found a significant increase in OPCs and immature oligodendrocytes (PDGFRa-positive) in mice at 3 weeks and 6 months of age (Fig. 3c). However, there were no changes overall in the total number of proliferating

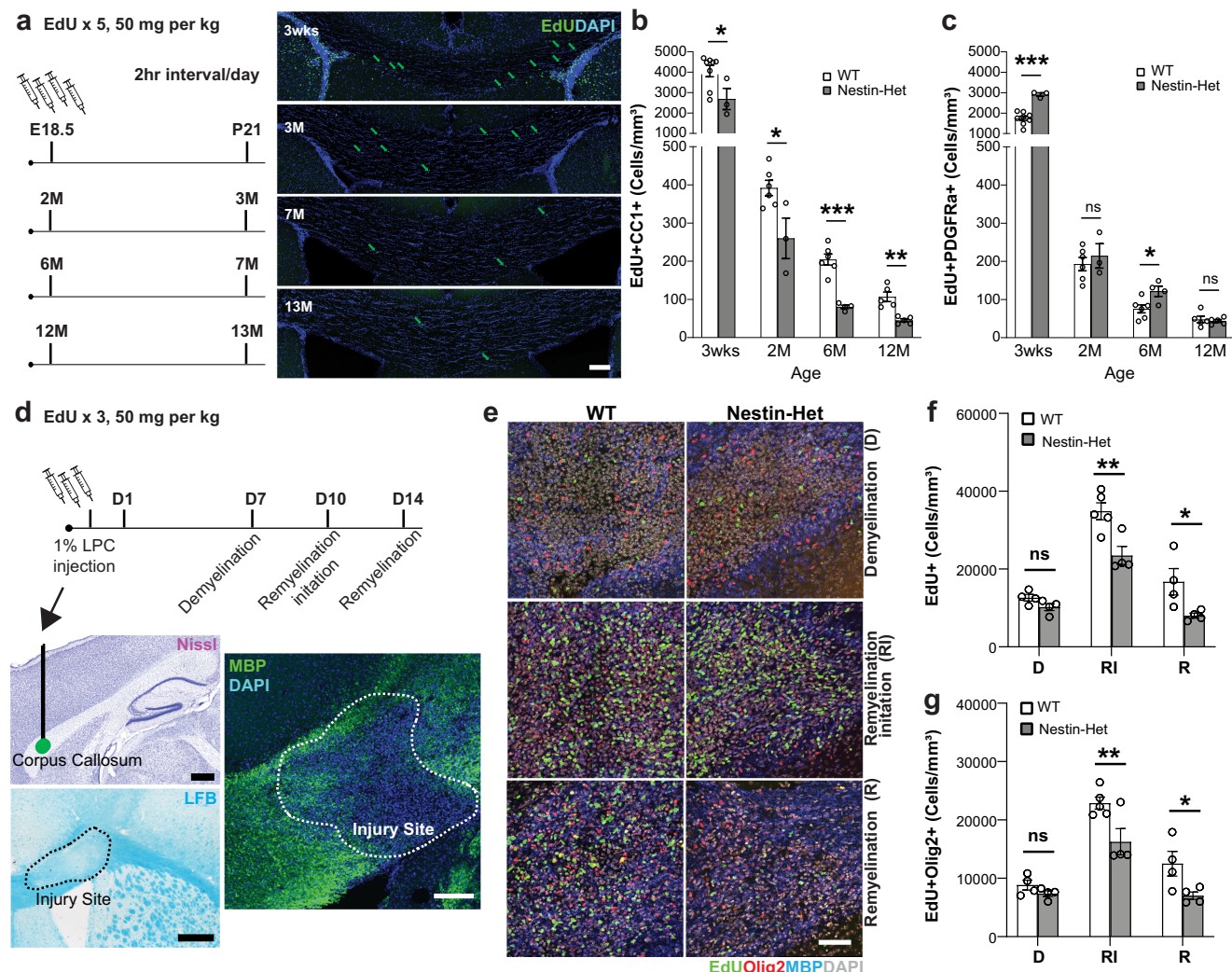

**Fig. 3 | Impaired oligodendrocyte maturation and motility in *Anks1b* Nestin-Het mice. a** To label newborn cells: pregnant females at E18.5 or mice at 2, 6, or 12 months of age were injected with EdU IP five times, in 2-h intervals. Four weeks after injections, mice were sacrificed and brain tissue was sectioned and immunostained to identify newborn cells (EdU+) and oligodendrocyte developmental stage (PDGFRa for OPCs and immature oligodendrocytes, CC1 for mature oligodendrocytes). Scale bar = 100 μm. **b** Significantly fewer mature oligodendrocytes were observed in *Anks1b* Nestin-Het mice, (T-test, two-tailed: 3weeks *p* = 0.035; 2 months *p* = 0.023; 6 months *p* = 0.0001478; 12 months *p* = 0.00158). **c** with concomitant increases or unchanged numbers of immature oligodendrocytes throughout all ages tested, (T-test, two-tailed: 3weeks *p* = 0.0001587; 6 months *p* = 0.0264). Number of animals at 3 weeks (WT: 8, Het: 3), 2 months (WT: 6, Het: 3), 6 months (WT: 6, Het: 4), 12 months (WT: 5, Het: 5), 4 sections per mouse. **d** (*Upper*)

For myelin lesion experiments: 1% LPC (lysophosphatidylcholine) was injected into callosal regions to lesion myelin sheaths. EdU was injected IP to track newborn cells, and mice were sacrificed 7 (Demyelination-D), 10 (Remyelination Initiation-RI), and 14 days later (Remyelination-R). (*Lower*) Lesions were observed as areas of decreased MBP expression or LFB staining. Scale bar = 20 μm. **e** Representative image showing newborn oligodendrocyte (EdU+ and Olig2+ together) accumulation in lesioned areas in both genotypes. Scale bar = 20 μm. **f** Quantitation showing that fewer cells overall (*n* = 4 WT and 4 Het mice for condition D, 5 WT and 4 Het mice for RI, and 4 WT and 4 Het mice for condition R), RI; *p* = 0.00056, R; *p* = 0.00722), and (**g**) fewer oligodendrocyte lineage cells accumulate in lesioned areas (*n* = same as (**f**), RI; *p* = 0.00247, R; *p* = 0.01314), suggesting impaired maturation or migration of OPCs. (**b, c, f, g**) Error bars reflect mean ± s.e.m. Source data for **b, c, f**, and **g** are provided as a Source Data file.

cells as measured by BrdU-positive cells in neurogenic subventricular or subgranular zones (Supplementary Fig. 3a, b). These results provide evidence for impaired maturation.

Because oligodendrocyte migration is important for white matter integrity, we performed a remyelination experiment[57] by injecting lysophosphatidylcholine (LPC) into the corpus callosum to generate a myelin lesion, and tracked OPCs in Nestin-Het and WT mice. Mice were sacrificed at 7 days after injection to document the extent of the myelin lesion, at 10 days after injection to measure the initial stages of remyelination, and at 14 days after injection when complete remyelination would be expected (Fig. 3d, e). Newborn oligodendrocyte-lineage cells were tracked using EdU as described above, and both myelin deposition and OPC numbers were assessed. As expected,

newborn oligodendrocytes accumulated at lesioned areas in WT mice (Supplementary Fig. 3c, d), and then dispersed throughout the corpus callosum (Supplementary Fig. 3e), validating the assay and timeline selected. However, we found significantly fewer newborn cells in demyelinated areas in Nestin-Het mice (Fig. 3f) compared to WT littermates. Moreover, significantly fewer of these cells were of oligodendrocyte lineage, as indicated by co-labeling with Olig2 (Fig. 3g). Interestingly, we did not observe a difference in the extent of MBP deposition at lesioned areas between genotypes (Supplementary Fig. 3f). These effects were observed during early remyelination (10 days post lesion) and persisted 2 weeks post lesion, suggesting that AIDA-1 plays an important role in the birth, maturation, and/or the migration of OPCs into demyelinated areas.

## AIDA-1 is expressed in oligodendrocytes

Deficits in myelination and oligodendrocyte numbers were unexpected given that *Anks1b*-encoded AIDA-1 is one of the most abundant proteins at neuronal synapses[4,5] and has no published role in CNS myelination. To explore whether cell-autonomous effects played a role in the observed phenotypes, we searched through single-cell transcriptomic databases and found that *Anks1b* is selectively expressed in the CNS not only in neurons, but also oligodendrocytes and OPCs[33,58,59] (Fig. 4a, Supplementary Fig. 4). This is consistent with recent studies suggesting the expression of AIDA-1 in these cells[30–33]. Immunocytochemistry using a knockdown-verified antibody against AIDA-1[3,60] in rat neuron-glia primary cultures shows that AIDA-1 is present in punctae throughout the cell body of oligodendrocytes (Fig. 4b). Fluorescent in situ hybridization (FISH) and RT-PCR confirms the presence of different *Anks1b* transcripts in these cells, as well as corpus callosum in mouse acute brain sagittal sections (Fig. 4c, Supplementary Fig. 5a, b). Western blots of mouse primary cell cultures revealed several AIDA-1 isoforms in oligodendrocytes, and at higher relative levels compared to neurons (Fig. 4d). No expression was observed in astrocytes (Fig. 4d), which corroborates diverse single-cell transcriptomic databases (Supplementary Fig. 4). Taken together, these results confirm that AIDA-1 is expressed in oligodendrocytes and OPCs.

## Loss of *Anks1b* from oligodendrocyte lineage cells results in oligodendrocyte dysfunction

To test the oligodendrocyte-specific contribution to the phenotypes observed, we conditionally deleted *Anks1b* from these cells using *Olig2*-Cre mice. Previously described *Anks1b* floxed mice[7] were crossed to *Olig2*-Cre mice (tm1.1(Cre), Jackson labs) on a similar C57BL/6J background. Oligodendrocyte-specific *Anks1b* heterozygous (Olig2-Het) and homozygous (Olig2-KO) oligodendrocyte-specific *Anks1b* knockout mice were viable, although knockout mice were born at rates below expected Mendelian ratios and were smaller than their WT counterparts (Fig. 5a). To confirm the specificity of Cre recombinase activity, we crossed *Olig2*-Cre mice to Rosa26-GFP reporter mice, and harvested brains were clarified using CUBIC[61] and imaged by light sheet microscopy (Fig. 5b). As would be expected for oligodendrocytes, GFP-expressing cells appeared tiled, small and with fine cloud-like processes (Fig. 5b, Supplementary Fig. 6a), and

overwhelmingly expressed in corpus callosum where they co-localized with MBP (Fig. 5c). These results corroborate the extensive use of *Olig2*-Cre lines to target oligodendrocytes[62–66], as well as studies that show that *Olig2* expression is highly specific and universal for oligodendrocyte lineage cells[67].

As in *Anks1b* Nestin-Het mice, imaging experiments show that Olig2-Het mice exhibit fewer oligodendrocytes in the corpus callosum (Fig. 5d), and Western blots confirm reduced MBP and decreased oligodendrocyte makers Olig2 and Sox10 overall (Fig. 5e, f). TEM imaging confirms a decrease in myelination of callosal axons as measured using g-ratios (Fig. 5g, h). To validate the haploinsufficiency of *Anks1b* in this model, we performed Western blots on lysates from brain tissues and primary cell cultures. Whole brain tissue lysates from WT and Olig2-Het mice revealed no discernible changes in AIDA-1 expression (Fig. 5i, j). Because heterozygosity in a small subset of cells would not be detectable by Western blot, this result is consistent with specific cre-recombination in OPCs and oligodendrocytes, which represent only 5–10% of all brain cells. However, AIDA-1 expression was substantially reduced in cultured primary oligodendrocytes, but not in neuronal cultures grown from Olig2-Het mice[68] (Fig. 5i, j). Together, these results validate the specificity of the *Olig2*-Cre driver for oligodendrocytes and suggest an oligodendrocyte-autonomous role for AIDA-1 in the phenotypes observed.

## Oligodendrocytes deficient in *Anks1b* display Rac1-based functional deficits

Western blots (Fig. 6a) and imaging analyses (Fig. 6j, k) show that oligodendrocytes from Olig2-Het mice have deficits in the expression of MBP, a marker of mature oligodendrocytes. Similar results were also observed in rat oligodendrocytes transduced with previously validated AIDA-1 specific shRNAs[7,60] (Fig. 6b, c; Supplementary Fig. 10c, d), suggesting that acute AIDA-1 knockdown can also lead to deficits in OPC maturation. Moreover, these results corroborate the assertion that myelination phenotypes observed in *Anks1b*-deficient mice originate from oligodendrocyte-autonomous deficits. To glean insights into the molecular mechanism linking AIDA-1 to cellular and behavioral phenotypes, we reanalyzed our published AIDA-1 interactome[3] and proteomic analyses of synaptosomes isolated from *Anks1b* Nestin-Het mice[36]. We reasoned that proteins that: 1) bind to AIDA-1 and 2) are

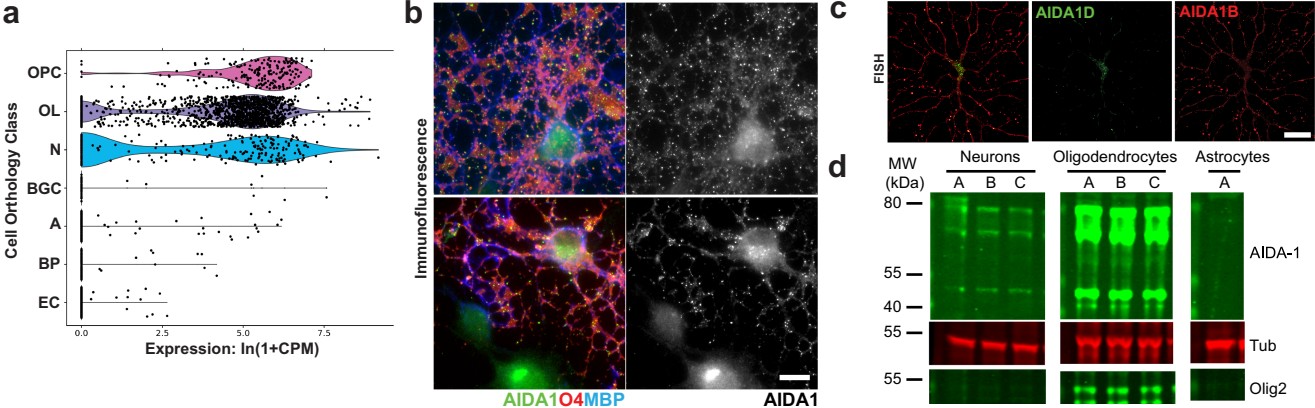

**Fig. 4 | AIDA-1 is expressed in oligodendrocyte lineage cells. a** Partial list of *Anks1b* transcript expression data among 236 cell types as documented in *Tabula Muris*. Substantial *Anks1b* expression is observed in only 3 cell types: OPC (oligodendrocyte precursor cells), OL (oligodendrocytes), and N (neurons). Also see Supplementary Fig. 4; BGC: Bergmann glial cell, A: astrocyte, BP: brain pericyte, EC: endothelial cell. **b** Immunocytochemistry image showing AIDA-1 is expressed in puncta throughout oligodendrocyte cell bodies. MBP: myelin basic protein, O4: Oligodendrocyte Marker O4. Scale bar = 10 µm. Representative image of >20 cells from 3 independent cell cultures. **c** Fluorescent in situ hybridization reveals

prominent expression of larger *Anks1b* transcripts encoding for large (AIDA-1B) and small (AIDA-1D) AIDA-1 splice variants throughout oligodendrocyte cell bodies. Scale bar = 10 µm. Representative image from 3 independent cell cultures.
**d** Western blot of lysates from neuronal, oligodendrocyte, and astrocyte primary cell cultures isolated from WT mouse pups (P1-P5), with each letter representing a separate mouse (*n* = 3). Several AIDA-1 splice variants (immunoreactive bands) are observed in oligodendrocytes and at higher levels compared to neurons. 10 µg of total protein lysate was loaded. No expression was observed in astrocytes. Tub: tubulin.

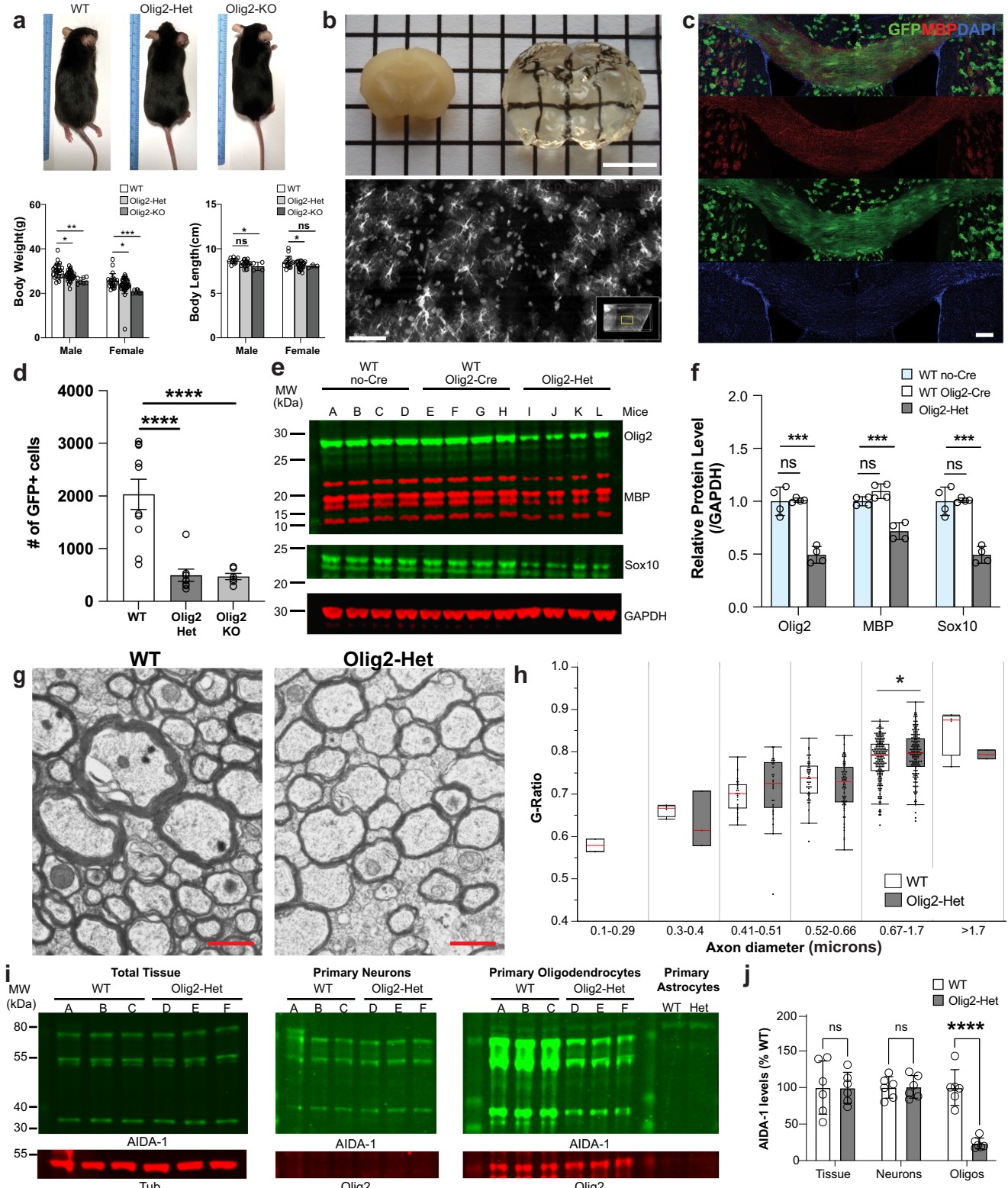

altered by its absence, would be strong candidates for functional links. A weighted interaction score based on co-immunoprecipitation data from several AIDA-1 antibodies normalized to a non-specific control revealed many small Rho GTPases and their regulators, as well as key proteins involved in myelination or oligodendrocyte function (Fig. 6d, Supplementary Fig. 7a). We then compared these proteins to our previous synaptic proteomic dataset, which identified proteins that were up- or down-regulated at hippocampal synapses in *Anks1b* Nestin-

Het mice (Fig. 6e, Supplementary Fig. 7b, c, Supplementary Data 2). An unbiased dataset of proteins that showed at least a 10% change compared to their respective controls in both proteomic studies were then included in a StringDB network model to identify functional clusters (Fig. 6e, f). Analyses revealed that by far the most enriched protein domains amongst this group were associated with GTPase catalytic activity (Fig. 6f). "Component" and "Function" Gene Ontology (GO) annotations similarly point to an enrichment for GTP binding as well as

**Fig. 5 | Generation and characterization of *Anks1b Olig2*-Cre transgenic mice.**
**a** *Anks1b* Olig2-Het and Olig2-KO mice were generated by crossing *Olig2*-Cre mice with *Anks1b* floxed mice. Olig2 transgenics were smaller and lighter than WT littermate controls (body weight *n* = 49 WT, 74 Olig2-Het, 14 Olig2-KO; body length *n* = 28 WT, 45 Olig2-Het, 7 Olig2-KO); Two-way ANOVA with Sidak's posthoc corrections. (Weight- Male: WT vs Het, *p* = 0.0126; WT vs KO, *p* = 0.0014; Female: WT vs Het, *p* = 0.0379; WT vs KO, *p* = 0.0006. Length- Male: WT vs KO, *p* = 0.0321; Female: WT vs Het, *p* = 0.016). **b** *Olig2*-Cre transgenic mice were crossed to GFP Cre-reporters (Rosa26-GFP), and, (*upper*) brain tissues were clarified using CUBIC and imaged by light sheet microscopy. Scale bar = 5 mm. (*Lower*) GFP-expressing cells were morphologically consistent with oligodendrocytes; scale bar = 20 μm.
**c** Immunocytochemistry performed on CUBIC clarified brains shows that GFP-labeled cells are enriched in corpus callosum and colocalize with MBP, confirming correct targeting of Cre recombinase to oligodendrocytes; scale bar = 100 μm.
**d** Quantitation of oligodendrocytes (GFP+ cells) in corpus callosum from acute coronal slices. As in *Anks1b* Nestin-Het mice, we found significantly fewer oligodendrocytes in *Anks1b* Olig2-Het and Olig2-KO mice (*n* = 8 WT, 6 Het, 4 KO, 8-10 slices from each mouse); One-way ANOVA with Sidak's posthoc corrections.
**e** Western blots of whole brain lysates from mice with indicated genotypes and for

markers of oligodendrocytes (Olig2 and Sox10) and MBP. Each lane of the Western blot represents a different mouse. **f** Results show decreased expression of oligodendrocytes markers and MBP expression, recapitulating results observed in Nestin-Het mice. (*n* = 4 mice per genotype). Mice expressing *Olig2*-Cre but not the floxed *Anks1b* allele (WT *Olig2*-Cre), had similar Olig2, MBP, and Sox10 expression compared to mice expressing the floxed allele alone (WT no-Cre); One-way ANOVA with Sidak's posthoc corrections. **g** Representative TEM images of callosal cross-sections from WT and Olig2-Het mice; scale bar = 1 μm. **h** G-ratio quantitation binned across the indicated axonal diameters shows significant myelination deficits in axons 0.67–1.7 μm in diameter (*n* = 3 WT, 3 Olig2-Het mice). Box plots show the 25th–75th quantiles (box), mean (red line), and whiskers delineate 3 standard deviations from the mean; T-tests, two-sided, *p* = 0.0206. **i** Western blots showing AIDA-1 expression in primary cell cultures isolated from WT (*n* = 3) and Olig2-Het pups (*n* = 3) (P1-P5). Significant decreases in AIDA-1 expression are seen in oligodendrocytes, but not neuronal cultures. Each lane of the Western blot represents a different mouse. **j** Quantitation of protein expression from (**i**) normalized to WT and tubulin (Tub). (*n* = 3 WT, 3 Olig2-Het mice).T-tests; two-tailed. **a, d, f, j** Error bars reflect mean ± s.e.m. **a, d, f, h, j** ***, ****p* < 0.0005, *p* < 0.00005. Source data for **a, d, f, h**, and **j** are provided as a Source Data file.

links to the myelin sheath (Fig. 6f). The prominence of Rac1 and associated regulators in these analyses confirmed our previous research pointing to Rac1 dysfunction in mouse models of ASD[36].

To directly test the effects of AIDA-1 on Rac1 function, we used a cutting-edge Fluorescence Resonance Energy Transfer (FRET)-based Rac1 biosensor to measure Rac1 activity[69,70]. To perform live imaging of primary oligodendrocytes (Fig. 6g), OPCs from Olig2-Het and WT mice were grown as described and then transduced with viruses to express the Rac1 biosensor. Two days after transduction, oligodendrocytes were imaged for 16 minutes in the presence of a Rac1 stimulator (CN04, Cytoskeleton, inc; IGF-1) to increase basal activity (Fig. 6h, i). While stimulation showed minimal effects on the overall FRET signal in this time frame, cells derived from Olig2-Het mice showed a marked increase in Rac1 activity compared to cells from WT mice (Fig. 6i). Similar experiments performed in rat oligodendrocytes transduced with AIDA-1 shRNAs also yielded significant differences in Rac1 activity, but curiously in the opposite direction from mouse cells (Supplementary Fig. 10a, b). To test whether abnormal Rac1 activity was linked to the maturation deficits observed in cells, we treated Olig2-Het oligodendrocytes for 5 days with Rac1 activator and found that this rescued the expression of MBP (Fig. 6j, k). These results corroborate the hypothesis that abnormal Rac1 activity underlies the myelination phenotypes observed in Olig2-Het mice.

### Loss of *Anks1b* from oligodendrocyte lineage cells results in behavioral correlates of ASD

To test the contribution of oligodendrocyte dysfunction to ANDS, we carried out behavioral assays in *Olig2*-Cre *Anks1b* conditional knockout mice across domains relevant to clinical features of this genetic disease. As in the Nestin-Het mice, no deficits were observed in novel object placement (Fig. 7a), a test of hippocampus-dependent learning and perseverative behaviors. Because we had found reduced anxiety-like behavior in Nestin-Het mice[3], we examined the behavior of *Olig2*-Cre transgenic mice on the elevated plus maze. Results were mixed as Olig2-Het and Olig2-KO mice spent more time in the open arms of the elevated plus maze (decreased anxiety) (Fig. 7b) but covered less distance in the open arms (increased anxiety) (Supplementary Fig. 8d). There were no differences in center exploration in the open-field test or in the forced swim test, two other measures of anxiety-like behaviors (Supplementary Fig. 8b, c). Hyperactivity present in Nestin-Het mice[3] was not observed in *Olig2*-Cre transgenic mice, as these showed no changes in locomotor activity in the open field (Supplementary Fig. 8a, c).

Autism is a prominent feature of ANDS and Nestin-Het mice have deficits in sensory reactivity and impaired social behaviors[3]. Like Nestin-Het mice, Olig2-Het and Olig2-KO mice of both sexes show significantly increased sensory hyperreactivity as demonstrated by increased startle to sounds at 100 db and 110 db, but not at 90db compared to WT littermates (Fig. 7c). Unlike Nestin-Het mice, *Olig2*-Cre transgenic mice had no impairment in sensorimotor gating as tested using pre-pulse inhibition (Supplementary Fig. 8e). Like Nestin-Het mice, Olig2 mice showed no differences in stereotypic and perseverative behaviors such as grooming (Supplementary Fig. 8a). Surprisingly, we found that both Olig2-Het and Olig2-KO mice displayed significantly decreased social preference in the three-chamber assay (Fig. 7d). For comparison, we also analyzed mice with homozygous *Anks1b* deletion specifically from forebrain excitatory neurons, using *Camk2a*-Cre (Camk2a-KO; generated previously[7]), or from cerebellar Purkinje neurons using *L7*-Cre (L7-KO). These neuronal subtypes display high *Anks1b* expression as observed in the Allen Brain Atlas (Supplementary Fig. 8g). Notably, only Olig2-Het and Olig2-KO mice displayed deficits in social preference similar to Nestin-Het mice (Fig. 7d), showing that social deficits in a mouse model of ANDS can originate from targeted oligodendrocyte disruption. A characterization of Camk2a-KO mice (Supplementary Fig. 9a) revealed that they did not exhibit any changes in the area of the corpus callosum compared to WT controls (Supplementary Fig. 9b), or changes in overall MBP expression (Supplementary Fig. 9c), supporting a correlation between deficits in social behaviors and white matter abnormalities. Overall, no significant sex differences were observed in any behaviors evaluated. Statistical analyses for the Olig2-Het (*n* = 82), Olig2-KO (*n* = 13), and WT (*n* = 53) mice demonstrate robust results with high power (beta > 0.85) for all tests where results were significant and beta >0.95 for social preference tests.

### Clemastine rescues social deficits in mice lacking *Anks1b* in oligodendrocytes.
To test whether the social impairments were caused by the myelination deficits, we sought compounds shown to induce myelination. Conveniently, we found that clemastine fumarate is a well-tolerated, FDA approved, and blood brain barrier permeable compound shown to increase myelination specifically by increasing OPC maturation[17,35,71–73]. We first tested this compound in primary oligodendrocytes and found that long-term treatment (5 days, 1 μM clemastine) significantly increased MBP expression in cells derived from Olig2-Het mice (Fig. 8a, b), as well as in rat oligodendrocytes transduced with shRNAs to knock down AIDA-1 (Supplementary Fig. 10f). We then took Olig2-Het and WT mice aged 6–7 months that had been previously tested for social preference (at 3–4 months of age) and injected them

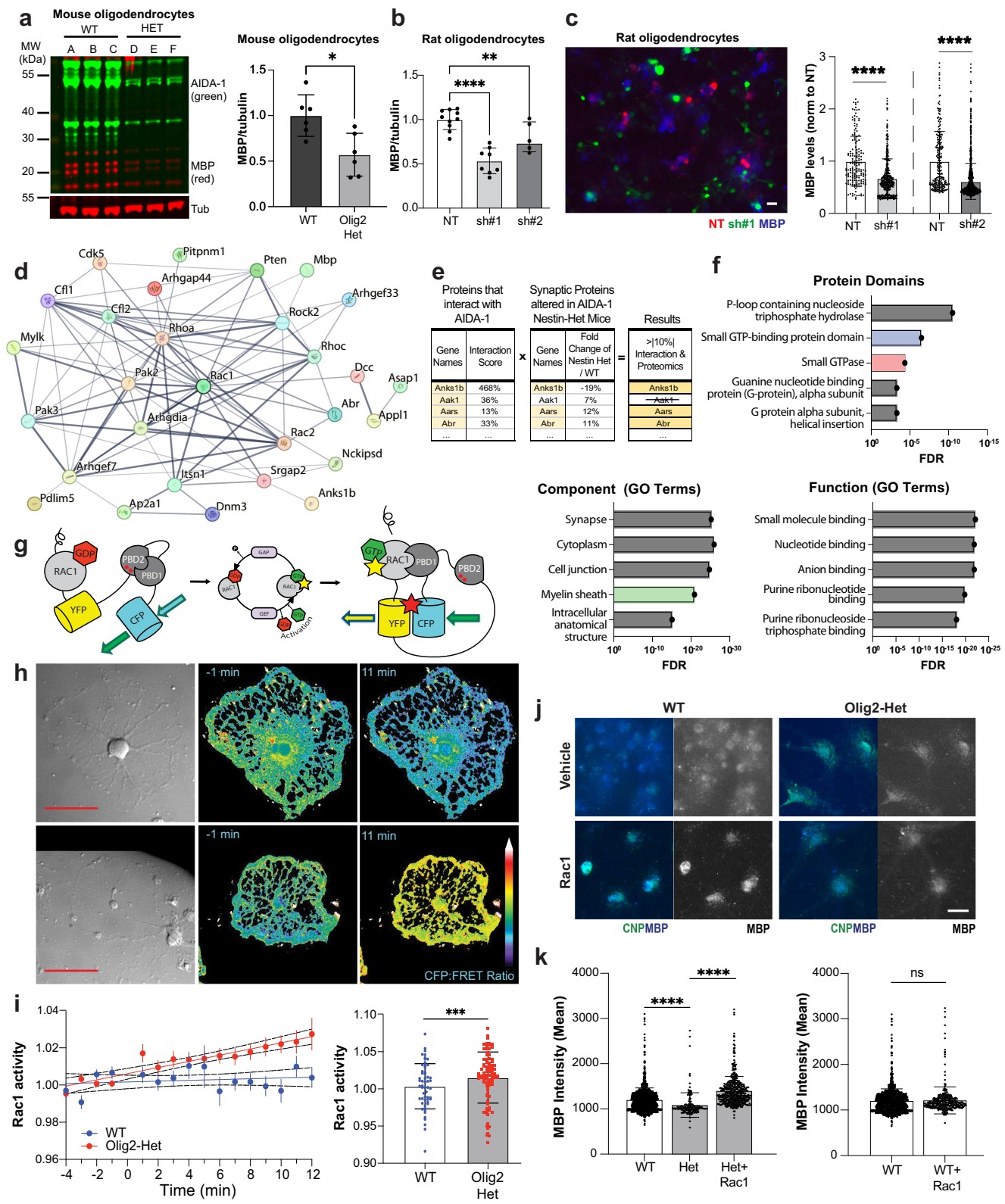

with clemastine (10 mg/kg) or vehicle daily for 14 days IP and allowed them to recover for 7 days (Fig. 8c). Clemastine had no effects on overall locomotor activity (Fig. 8d). Notably, paired results show that clemastine treatment significantly rescued social preference in mice that had previously exhibited no social preference (Fig. 8e). Drug effects were largest for mice that initially displayed the poorest scores. No effects were observed for Olig2-Het mice treated with vehicle (Fig. 8f), or WT controls treated with either clemastine or vehicle (Fig. 8g), showing that

improved social preference was neither a consequence of behavioral retesting, nor of vehicle injections. Following behavioral testing, a randomized subset of the mice was euthanized (Olig2-Het clemastine-treated $n = 3$, Olig2-Het vehicle-treated $n = 3$), and dissected brains were fixed and prepared for TEM analyses (Fig. 8h). We found that clemastine treatment resulted in a significant decrease in g-ratio across most callosal axon diameters, which indicates a substantial increase in myelination (Fig. 8i). These results show that OPC maturation and induction

**Fig. 6 | Deficits in Rac1 activity underlie abnormal oligodendrocyte function. a** Western blots of lysates from oligodendrocyte monocultures obtained from WT and Olig2-Het mice (*n* = 3 mice per genotype, 2 technical replicates) show reduced MBP expression; T-test, two-sided. *p* = 0.00931. Primary rat oligodendrocyte culture (**b**) (lysates; NT: *n* = 6 independent cultures, 10 lanes; sh#1: *n* = 4 independent cultures, 8 lanes; sh#2: *n* = 3 independent cultures, 6 lanes: One-way ANOVA, NT vs sh#1, *p* = 0.0041) and (**c**) imaging (*n* = 4 independent cultures for each group; NT1 (164), sh#1 (389), NT2 (238), sh#2 (1167) cells) likewise show reduced MBP expression after knockdown of AIDA-1 with both shRNA viruses (sh#1 and sh#2) compared to non-targeting control virus (NT). One-way ANOVA with Dunnett's posthoc corrections; scale bar = 20 μm. NT vs sh#1: *p* = <0.0001, NT vs sh#2: *p* = 0.0041. **a** Normalized to tubulin (Tub) and (**a–c**) to WT/NT as appropriate. **d** A network map generated by StringDB showing the subset of proteins from the interactome data associated with small GTPases. **e** A sample of the cross-referenced interactome data with the previously published synaptic proteomics. Proteins that scored >|10%| fold-change over control in both the interactome and the proteomics were selected (Full list in Supplementary Data 2) were selected as strong functional links. **f** An analysis of protein domains found in this subset of proteins, shows that

these were overwhelmingly associated with small GTPase function. Gene ontology (GO) terms of this protein subset indicated links to myelin and oligodendrocyte related functions. FDR: False Discovery Rate. **g** Schematic of the Rac1 (FRET: Fluorescence Resonance Energy Transfer)-based biosensor. PBD: p21-binding domain of Pak1. **h** Representative mouse cells expressing the G-protein biosensor and imaged for FRET. Scale bar = 50 μm. **i** (*Left*) Plot showing the Rac1 activity observed for WT and Olig2-Het cells, represented by the CFP to FRET ratio over time, with x-axis min 0-12 representing post stimulation imaging. (Right) bar graphs showing minutes 7-12 grouped together. *n* = 50 WT and 90 Het cells from 12 mice (5 WT and 7 Het mice); T-test, two-sided. CFP: Cerulean Fluorescence Protein. **j** Representative image of WT and Olig2-Het oligodendrocyte monocultures immunostained for MBP and the mature oligodendrocyte marker CNP. Scale bar = 20 μm. **k** (*Left*) Rac1 stimulation restores MBP expression to above-WT levels in CNP-positive oligodendrocytes and (*right*) had no effect on WT cells; T-test; two-sided; *n* = 4 independent cultures for each group; WT (1268), Het (147), WT+Rac (349), Het+Rac (452) cells. WT vs Het: *p* = 0.000064, Het vs Het+RAC: *p* = 4e−7. MBP: myelin basic protein, CNP: 2′,3′-cyclic nucleotide 3′-phosphodiesterase.

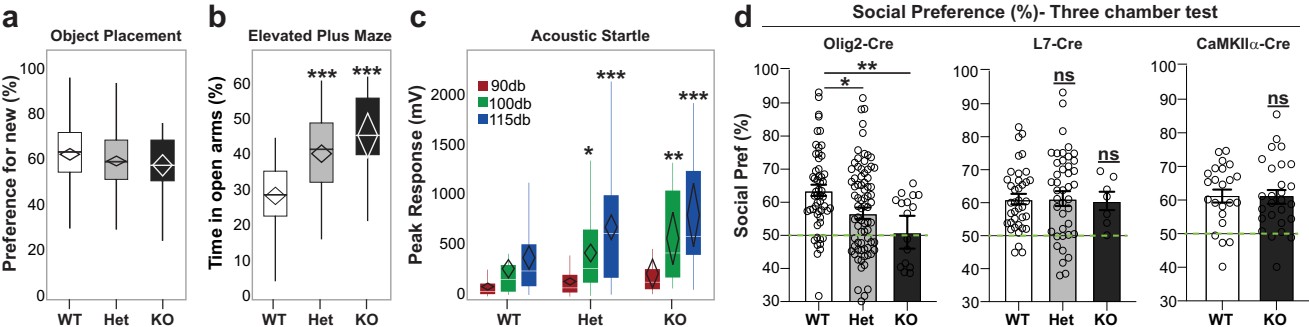

**Fig. 7 | Mice with oligodendrocyte specific *Anks1b* deletion display behavioral deficits associated with ASD. a** Neither Olig2-Het nor Olig2-KO mice show learning deficits in the object placement test, a hippocampus-dependent memory assay. All 3 groups averaged above a passing score (>50% preference for new location) using a 40-minute retention interval. **b** Olig2-Het and Olig2-KO mice showed a robust reduction in avoidance behaviors compared to WT controls, spending more time in the open arms as a percentage of total time in open and closed arms of an elevated plus maze (but see Supplementary Fig. 8d). One-way ANOVA. **c** Peak magnitude of the acoustic startle reflex is robustly increased in Olig2-Het and Olig2-KO mice at 100db and 115db. (\**p* = 0.0334, \*\**p* = 0.00281) **d** In the three-chamber test, (*left*) Olig2-Het and Olig2-KO mice show significantly reduced and borderline preference for a conspecific mouse over an inanimate object (One-way ANOVA, Dunnett's post hoc, WT vs Het: *p* = 0.0174; WT vs KO: *p* = 0.0077). This reduction in social preference was not observed in mice with *Anks1b* deletion from (*middle*) cerebellar

Purkinje cells (L7-Het and L7-KO) or (*right*) forebrain specific excitatory neurons (Camk2a-KO). Dotted green line indicates 50% preference. **a–c** Box plots show 25th–75th quantiles (box), median (black line), 95% confidence intervals (diamond), and range (black whiskers). Bar graphs in **d** show mean ± s.e.m. with all data points shown. For *Anks1b* Olig2-Het and Olig2-KO experiments, *n* = 53 WT, 82 Het, 15 KO (two measurements made for object placement). For L7-Het and L7-KO experiments, *n* = 38 WT, 44 Het, 7 KO. For *Anks1b* Camk2a-KO experiments, *n* = 23 WT, 27 KO. No significant sex differences were observed in any behaviors evaluated. If 2-way ANOVA showed significant main effect of genotype, *post hoc* 2-sided Student's *t* test was performed, \*, \*\*, \*\*\**p* < 0.05, <0.005, <0.0005. All significant results show high power (b > 0.85). For social preference test in Olig2-Het and Olig2-KO statistical power b > 0.95. Source data for all panels are provided as a Source Data file.

of new myelin formation through clemastine treatment can improve social behaviors even in 6–7-month-old mice that are significantly beyond critical periods for myelination. Overall, these results strongly support the assertion that behavioral deficits in Olig2-Het mice are caused by deficits in oligodendrocyte function and myelination.

## Discussion

Here we reveal an unexpected role for *ANKS1B* in oligodendrocyte maturation and function, and a potential role for oligodendrocytes in sociability and NDDs. We show that ANDS patients display profound white matter abnormalities, and that CNS-wide (Nestin-Het) mouse models reveal similar white matter aberrations (Fig. 1), as well as decreases in oligodendrocyte abundance, maturation, and myelination in the corpus callosum (Figs. 2 and 3). The *Anks1b*-encoded protein AIDA-1 was previously reported to be neuron-specific[3,4,6,7,60,74] and expressed mainly at PSDs in dendritic spines[4–6,60]. Here we show that AIDA-1 is also expressed in oligodendrocyte lineage cells (Fig. 4). While expression of *Anks1b* in neurons and oligodendrocyte lineage cells points to multifactorial dysfunctions in disease etiology, selective

*Anks1b* deletion from oligodendrocytes recapitulates oligodendrocyte phenotypes (Fig. 5) as well as the social and sensory correlates of ASD observed in Nestin-Het mice (Fig. 6). Notably, these effects were not observed in transgenic mice targeting neuronal populations that highly express *Anks1b*. Overall, our results suggest that oligodendrocyte-autonomous *Anks1b* haploinsufficiency mediates the deficits in white matter microstructure, oligodendrocyte abundance and maturation, and social behaviors observed in *Anks1b*-deficient mice.

We present a potential mechanism linking AIDA-1 to oligodendrocyte function by regulating Rac1 activity. We previously found that AIDA-1 interacts with Rho family GTPases and associated regulatory proteins[3,36]. Signaling pathways mediated by Rho Family GTPases are known to play pivotal roles in orchestrating the dynamics of the actin cytoskeleton. These signaling mechanisms have also emerged as crucial regulators of morphology, maturation, and migratory capacities of oligodendrocyte lineage cells[40–44]. Disruptions in these finely tuned processes might contribute to the phenotypes associated with oligodendrocytes that we observed in mouse models of ANDS. This

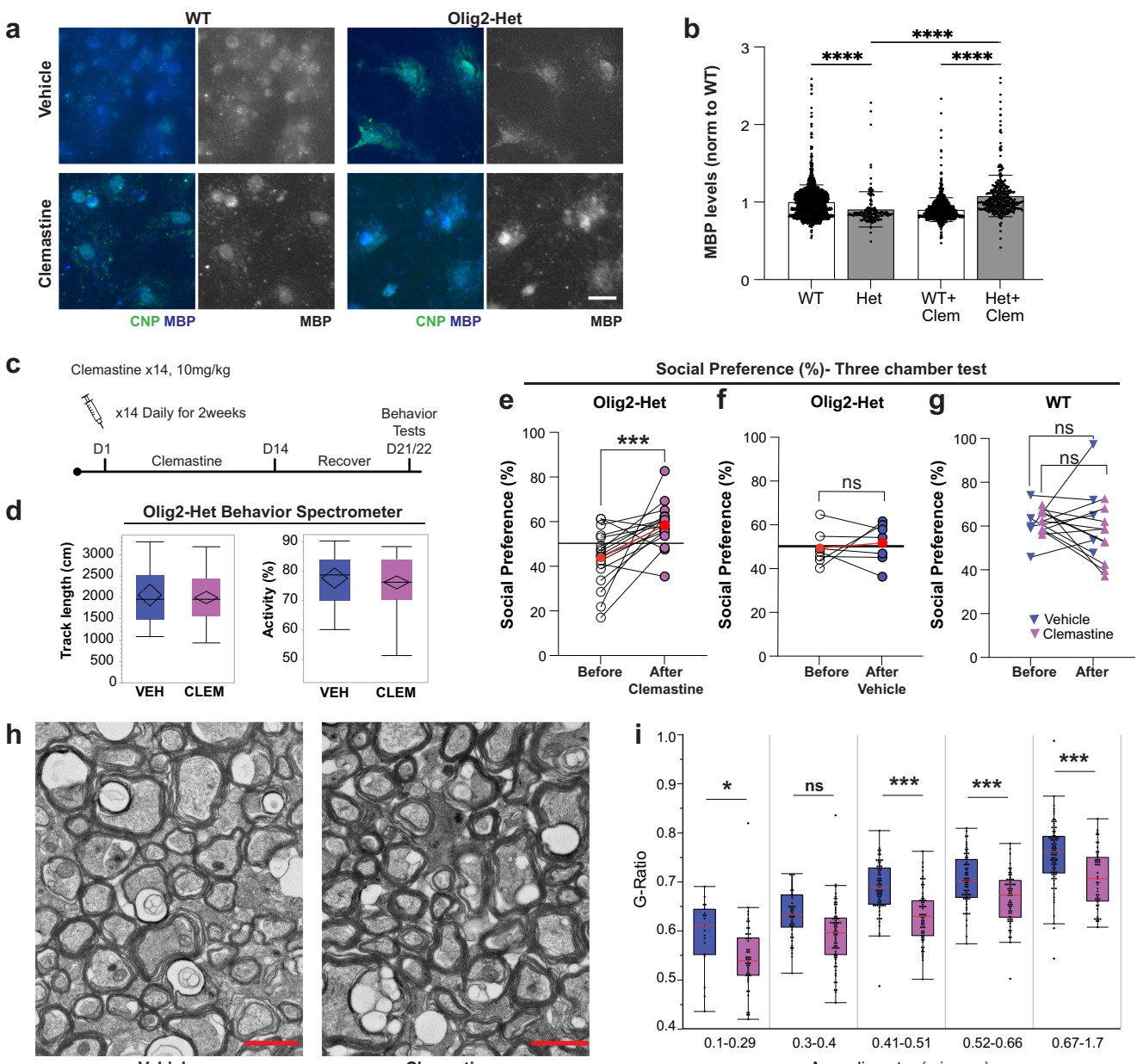

**Fig. 8 | Clemastine rescues deficits in social preference in *Anks1b* Olig2-Het mice. a** Representative image of WT and Olig2-Het oligodendrocyte mono-cultures (3 independent replicates) immunostained for MBP and the mature oligodendrocyte marker CNP. Scale bar = 20 μm. **b** Clemastine rescues deficits in MBP expression in CNP-positive primary oligodendrocytes cultured from Olig2-Het mice, but reduced expression in cells from WT mice; Two-way ANOVA with Sidak's multiple comparison; *n* = 4 independent cultures per group; WT (1268), Het(147), WT+Clem(776), Het+Clem(373) cells. Error bars reflect mean ± s.e.m. WT+Clem vs Het+Clem: *p* = 4e-7, Het vs Het+Clem: *p* = 4e-7. **c** Protocol for daily clemastine fumarate IP injections for 2 weeks. Mice were allowed to recover for 7 days before behavioral testing. Scale bar = 20 μm. **d** Total track length and activity as measured as time moving as a % of total time, showing that clemastine treatment had no overall effect on locomotion. Box plots show the 25th–75th quantiles (box), median (black line), 95% confidence intervals (black diamond), and range (black whiskers).

*n* = 39 mice. **e** Paired results show Olig2-Het mice performance in three-chamber social preference test before and after clemastine treatment. *n* = 18 mice. Paired T-tests. *p* = 0.00169. **f** Paired results from Olig2-Het mice treated with vehicle. *n* = 7 mice. Paired T-tests. **e**, **f** Red circles represent mean performance for each population and reference line indicates 50% preference. **g** Paired results from WT mice treated with clemastine or vehicle. (*n* = 5 mice (WT + Veh), *n* = 9 mice (WT + Clem); Paired T-tests, two-sided. **h** Representative TEM images of callosal cross-sections from Olig2-Het mice that were treated with either clemastine or vehicle; scale bar = 1 μm. **i** G-ratio quantitation binned across the indicated axonal diameters shows significant myelination deficits in axons 0.1–0.29 and 0.41–1.7 μm in diameter. Results are from 6 animals (3 per treatment condition). Box plots show the 25th–75th quantiles (box), mean (red line), and whiskers delineate 3 standard deviations from the mean: T-tests, two-sided. (**b**, **e**–**g**, **i**) *, ***, ****$p$ < 0.05, <0.005, <0.0005, <0.00005. Source data for **b**, **d**–**g**, and **i** are provided as a Source Data file.

suggests that the syndromic phenotypes we see caused by loss of *Anks1b* in oligodendrocytes may be caused by impaired interactions of AIDA-1 via Rac1 or Rac1-mediated processes. For example, p21-activated kinases (Pak1, Pak2, and Pak3) are highlighted in our proteomic and bioinformatic analyses (Fig. 6d–f). These processes are essential for proper oligodendrocyte development and differentiation,

with Pak1 promoting oligodendrocyte morphogenesis and myelination, Pak2 involved in proliferation and survival, and Pak3 involved in OPC differentiation[51,75–79]. Disruption of Rac1 and Pak1, 2 or 3 activities has been linked to various disorders, including ASD[78,80]. Other potential mechanistic pathways of interest include RhoA-Rock2, DCC, and Srgap2 (Fig. 6d–f). Our results show that AIDA-1 regulates Rac1 activity

in mouse oligodendrocytes (Fig. 6). Curiously, Rac1 activity in rat cells transduced with AIDA-1 shRNA was also significantly altered, but in the opposite direction (Supplementary Fig. 10a, b). The different results in mouse and rat cells could be due to species differences, developmental compensation in the Olig2-Het mice, or significant differences reported between the two types of cells[81]. Alternatively, these results could point to a dose-dependent effect on complex Rac1 functional dynamics, given that the amount of AIDA-1 knockdown observed in rat cells was small (Supplementary Fig. 10e) compared to the reduction observed in Olig2-Het mouse cells (Fig. 5i, j). In both cases, we were only able to reliably record Rac1 activity for 12 min post-stimulation because primary oligodendrocytes are highly sensitive and do not survive long in the live-cell FRET recording conditions. This means that we may have missed the complex temporal patterns displayed by RhoA family GTPases. We also report that long-term stimulation of Rac1 was able to rescue the expression of MBP in cells in culture (Fig. 6j, k). Taken together, these results suggest that the regulation of Rac1 is part of the mechanism underlying the interaction of AIDA-1 in oligodendrocyte and myelination processes, and that disruption of this mechanism is related to prominent phenotypes seen in ANDS patients.

It is worth noting that *Anks1b* (as EB1) was originally identified as a transcript upregulated in pre-B myeloid leukemia cells[82], that it is ectopically expressed in transformed cell lines[82,83], and has been linked to diverse cancers in GWAS studies[84,85]. While this research suggests a potential function in contact inhibition or cellular division, we did not detect any effects on overall cell birth in neurogenic zones of the CNS (Supplementary Fig. 3b). Despite the deficits in OPC maturation and significant reduction in the number of newborn oligodendrocytes in acutely demyelinated areas of Nestin-Het mice, we found that MBP was deposited at similar rates in lesioned areas compared to WT controls (Fig. 3). These results are consistent with recent findings showing that mature oligodendrocytes, and not OPC turnover, contribute to the regulation of myelination[86]. Another potential explanation for our findings is that our time points were insufficient to capture the contribution of newborn cells to remyelination. Because OPCs require at least two weeks (the time of our remyelinating assays) to differentiate into myelinating oligodendrocytes[17,18] it is possible that our assay quantified MBP deposited by pre-existing mature or immature oligodendrocytes present or near lesioned areas. Yet another explanation is that oligodendrocytes are highly heterogeneous[30] and *Anks1b* may not be expressed in cells that participate in acute remyelinating processes.

Studies of ASD etiology have implicated diverse convergent cellular pathways, including those regulating synaptic function[87–100]. Databases for ASD-linked genes such as SFARI Gene (Simons Foundation) are enriched in synaptic components, and mouse models for syndromic ASD are primarily associated with synaptopathies arising from mutations in highly synaptic proteins such as SHANKs, neurexins, neuroligins, CNTNAP2, and SynGAP1[95,101,102]. Given that *Anks1b*-encoded AIDA-1 is one of the most abundant proteins at neuronal excitatory synapses[60] and regulates NMDAR function[7], we expected that neuron-specific *Anks1b* deletion would drive ASD-related behavioral deficits. We focused on forebrain excitatory neurons (*Camk2a*-Cre) and cerebellar Purkinje cells (*L7*-Cre) because these cells most prominently express *Anks1b*, and because previous studies have linked these cell populations to other mouse synaptopathy models of ASD[103,104]. Surprisingly, only *Olig2*-Cre driven *Anks1b* conditional mutant mice had social deficits, suggesting a significant contribution of oligodendrocyte dysfunction to autism-like features in ANDS. Clearly, our experiments do not rule out other neuronal populations, early developmental processes in neurons, or interactions between oligodendrocytes and neurons in which AIDA-1 may participate. Moreover, while *Olig2*-Cre mice have been used extensively for the study of oligodendrocyte function[62–66], Olig2 expression has been reported in interneuron subpopulations early in development[105] as well as in

cerebellar Purkinje cells[106]. However, we saw no changes in established markers of interneurons (PV, SST) or cerebellar Purkinje cells (calbindin) throughout the brain (Supplementary Fig. 6b) and no changes in cerebellar Purkinje cell numbers in Nestin-Het mice (Supplementary Fig. 6c). Additionally, Purkinje-specific *Anks1b* knockout mice (L7-KO) exhibit no deficits in social preference (Fig. 7d), diminishing concerns that confounding expression of Olig2 in these cells play a role in the social deficits observed. However, we recognize that a limitation of our work is that we cannot fully rule out the influence of a small subset of interneurons on behavioral deficits in the Olig2-Het mice.

The important role for AIDA-1 at neuronal synapses[3,6,7,60] raises the possibility for a similar role in oligodendrocyte lineage cells. Seminal findings show that OPCs can also form functional synapses with neurons in the hippocampus[107–112], and that these synapses display properties similar to canonical neuronal synapses[113]. OPCs express components of PSDs[33,114], and ultrastructural images show PSD-like structures in OPCs and synaptic vesicles in presynaptic neurons[107,115,116]. Indeed, OPCs functionally express the full gamut of ionotropic and metabotropic neurotransmitter receptors, including AMPA, kainate, and NMDA receptors[109,110]. Well-established findings show that neuronal activity regulates oligodendrocyte proliferation, maturation, and myelination[117–121]. Similar to what has been observed in neurons[6], AIDA-1 may relay fast synaptic information from neuron-OPC synapses into the nucleus to regulate protein synthesis or other genetic signals necessary for OPC maturation. Punctate staining of AIDA-1 throughout cell bodies of oligodendrocytes (Fig. 4) may mark neuron-OPC synaptic junctions. Thus, an additional synaptopathy may underlie *ANKS1B* syndrome, one of unique neuron-OPC synapses as well as of canonical neuron-neuron synapses. Additional experiments will be required to elucidate how AIDA-1 regulates oligodendrocyte maturation and function.

Our comprehensive behavioral assessment of *Olig2*-Cre driven *Anks1b* conditional knockout mice reveals that some ASD-related behavioral phenotypes including social preference can originate from oligodendrocyte dysregulation. Together with a recent report of oligodendrocyte dysfunction in a rodent *Chd8* transgenic rodent model of autism[29], our results highlight an underappreciated role for these cells in NDDs. How these cells contribute to complex behaviors such as sociability is unclear. While tantalizing research suggests that oligodendrocytes can fire action potentials[110,122] and establish functionally connected circuits throughout the brain[123], any effects on myelination will clearly impact neuronal circuit activity. Changing myelinated axon structure by altering internode number and length, and myelin thickness and stability, will regulate conduction velocities to alter neural networks and circuit plasticity[117,124–126]. Two recent studies on memory consolidation showed that inhibiting active myelination through pharmacological or genetic manipulation blocks recall of spatial learning in both water maze and conditioned fear assays[17,18], and chronic administration of clemastine preserved the retrieval of remote fear memory[17]. While our results suggest that *Ansk1b* broadly regulates myelination by affecting oligodendrocyte maturation, more detailed analyses may reveal preferential targeting of specific circuits involved in regulating social behaviors. Targeted *Anks1b* expression in a specific oligodendrocyte subtype may provide the basis for selective regulation of neural circuits.

The few available therapies for NDDs include behavioral therapy, physical rehabilitation, and symptoms-based pharmacotherapy, but specific and effective treatments are lacking due to a poor understanding of disease mechanisms[127]. We found that clemastine fumarate rescues the impaired social preference in Olig2-Het mice. Recent high-throughput screens identified clemastine fumarate as possible therapeutic compounds for multiple sclerosis[35,128]. Clemastine is a blood brain barrier permeable compound approved by the FDA to treat seasonal allergies. It binds to the histamine receptor 1 but promotes OPC maturation and myelination by stimulating M1 acetylcholine

receptors[128]. Clemastine has shown promising therapeutic potential by promoting myelination in hypoxia, mouse models of demyelination, and neurodegenerative disorders[71,72,128,129]. It has recently been shown to relieve the severity of clinical symptoms from inflammatory demyelination[73,130] by increasing OPC maturation and myelination[17,35,71–73], as well as rescue deficits in memory consolidation due to abnormal myelination[17]. To our knowledge, our study represents the first use of clemastine to rescue phenotypes in an animal model of ASD.

Our research supports the notion that impaired myelination contributes to the pathogenesis of NDDs and directly implicates dysfunction of oligodendrocyte lineage cells in disease. Recent studies[26,131–136] corroborate long-standing evidence of deficits in white matter organization in children and adults with ASD[22–25]. In addition, analyses of differentially expressed genes in several mouse models of syndromic ASD highlighted dysregulation of oligodendrocyte function[136]. Although the developmental component of genetic diseases like ANDS has previously given little hope for therapeutic intervention in adults[137], new findings, including ours, indicate that at least some behavioral correlates of NDDs are reversible well into adulthood in animal models. This may be especially true for phenotypes caused by deficits in myelination, given that myelination is a dynamic process that remains plastic throughout the lifespan. In a SHANK3 model of ASD, morphological, functional, and behavioral anomalies such as self-injurious grooming and social interaction deficits (but not anxiety or motor deficits) are reversible in adult mice[128]. Here we find that deficits in social preferences can be ameliorated even in 7-month-old mice, an age that is significantly far into adulthood in these animals. Our results suggest that interventions targeting myelination may represent a potential therapy for ASD-related phenotypes in ANDS, and perhaps for other syndromic ASDs that display impaired myelination.

## Methods
### Reagents
Antibodies, reagents, mouse lines, and genotyping primers are detailed in Supplementary Data 3.

### Brain imaging
**Human subjects and clinical data.** All procedures were performed under the ethical approval by the Institutional Review Board (IRB) at AECOM and are described in IRB protocol #2011-320. Written consent was obtained from all participants prior to MRI and DTI analyses. The two affected individuals are men, and the neurotypical group was comprised of 50 men and 51 women.

**Patient MRI scans.** Imaging data was collected using a 3.0 T Philips Achieva TX scanner (Philips Medical Systems, Best, The Netherlands) with a 32-channel head coil. Structural 3D T1-weighted (T1W) magnetization-prepared rapid acquisition of gradient echo imaging was performed with TR/TE/TI = 8.5/3.9/900 ms, α = 8°, SENSE factor 2.0/2.6 in SI/RL directions, 1 mm$^3$ isotropic resolution, 240 × 240 × 220 matrix. Diffusion tensor imaging (DTI) data was acquired using 2D single-shot spin echo EPI with 32 diffusion-weighting directions at b = 800 s/mm$^2$ and TE = 56 ms, TR = 7.6 s, SENSE factor = 2.9, 2 mm$^3$ isotropic resolution, 128 × 128 matrix, 70 slices. An auxiliary field map scan was acquired using FOV = 256 mm, 4.0 mm$^3$ isotropic resolution, TR = 26 ms, TE/DTE = 2.5/2.3 ms, α = 26°. The field map is used to correct susceptibility-induced distortions in echo-planar (diffusion) scan.

**MRI data analysis.** Most of the image analysis steps, correction of DTI data for motion, eddy currents, EPI distortion as well as generation of map of fractional anisotropy and its rigid body registration to subject's T1W image, were performed using FMRIB-FSL package[138] as previously described[139]. Normative data from the Einstein Lifespan Study was pre-processed in a similar way and registered to the subject's T1W image using non-linear ANTs registration algorithm[140] for voxel-wise comparison using EZ-Map[141]. This subject-based method was shown to outperform template-based approach[54].

**In vivo mouse brain MRI.** All MRI data were acquired in a 9.4 T Varian Direct Drive system (Agilent Technologies, Santa Clara, CA, USA) at the Einstein Gruss Magnetic Resonance Research Center. A 14-mm diameter receive-only surface RF coil (Doty Scientific, Columbia, SC, USA) along with a 7-cm ID $^1$H transmit and receive body coil (M2M Imaging, Cleveland, OH, USA) were used. Anesthesia was achieved with ~1.25% isoflurane mixed with room air. Respiration was monitored with a pressure pad (SA Instruments, Stony Brook, NY, USA). Rectal temperature was maintained at ~38 C using warm air with feedback from a rectally placed thermocouple (SA Instruments). T2-weighted images were acquired using fast spin echo (fse) sequence covering the whole brain with the following parameters: TR = 3000 ms, ESP = 8.8 ms, segments = 48, ETL = 4, kzero = 3, effective TE = 26.4 ms, NSA = 4, data matrix = 192 × 192 (zero filled to 256 × 256), FOV = 25 mm$^2$, slices = 60, thickness = 0.3 mm, gap = 0 mm. Diffusion tensor image (DTI) was acquired with the following parameters: TR = 3000 ms, TE = 19.62 ms, kzero = 6, shots = 8, number of signals averaged (NSA) = 1, slices = 24, slice thickness = 0.5 mm, gap = 0, field of view (FOV) = 25 mm$^2$, data matrix = 128 × 128 (zero-filled to 256 × 256), b-value = 825.9 s/mm$^2$, G = 0.45 T/m, d = 2.56 ms, D = 8 ms, 42 directions plus 6 unweighted images, acquisition time = 40 min 18 s. 3D multiecho gradient echo (MGRE) images from which T2* mapping was generated were acquired with the following parameters: TR = 90 ms, 20 echoes with first echo at 1.78 ms, time between echoes 2.59 ms, flip angle 17, resolution = 125 × 125 × 125 μm$^3$, 0 with outer volume suppression.

**Ex vivo mouse brain MRI.** Fixed mouse brains were positioned in a plastic tube filled with Fomblin Y (perfluoropolyether; Sigma-Aldrich Co., St. Louis, MO). Imaging was performed on the same system and coils as described above. Parameters for fse sequence were as following: TR = 5000 ms, ESP = 17.54 ms, segments = 64, ETL = 4, kzero = 3, effective TE = 52.61 ms, NSA = 16, data matrix = 256 × 256, FOV = 12.8 mm$^2$, slices = 50, thickness = 0.3 mm, gap = 0 mm. DTI (30 orthogonal directions[142]) was accomplished using b-values of 939 s/mm$^2$, a 128 × 128 matrix and FOV of 12.8 mm$^2$, and 30 slices (0.3-mm thickness and no gap).

**MRI data analysis.** The Dorr mouse brain atlas[143] was registered to T2-weighted images covering the whole brain using ANTs[140] registration software. Maps of diffusion metrics including fractional anisotropy (FA), mean diffusivity (MD), axial diffusivity (AD), and radial diffusivity (RD) were generated using the FMRIB FSL diffusion toolbox[138]. Volumes of segmented regions[143] and average diffusion metrics over the segmented regions were then calculated.

***Anks1b* mouse models.** All protocols were approved by Albert Einstein College of Medicine's Institutional Animal Care and Use Committee (IACUC) in accordance with National Institutes of Health guidelines. Mice were maintained in a pathogen-free animal room under a 12-h light/dark cycle. Food and water *ad libitum* were available prior to the beginning of all experiments. Both sexes were used in biochemical, morphological, and behavioral studies. *Anks1b* floxed mice were generated as previously described[7]. All mice used for conditional knockdown are described in Supplementary Data 3. *Anks1b* Olig2-Het and Olig2-KO mice were generated by crossing previously described *Anks1b* floxed mice[7] with *Olig2*-Cre mice from Jackson Laboratories. Wildtype animals (WT) used throughout refer to one of these three lines: 1) animals with the Cre transgene and without the floxed Anks1b allele, 2) mice expressing the floxed allele without the Cre transgene, or 3) mice expressing neither transgene nor floxed allele.

**Tissue preparation, histology, clearing, immunofluorescence.** Animals were perfused with PBS followed by 4% paraformaldehyde (PFA). Brains were harvested and post-fixation was performed in 4% PFA at 4 °C for 24 h. Fixed brains were then equilibrated in 30% sucrose for 2 days. Entire brain sections (coronal 40 μm thickness) running anterior to posterior were collected using a microtome in serial order for histological analysis and immunolabeling analyses.

**Histology.** Nissl staining was performed using Cresyl Violet dye to measure the surface area of each region of interest. Briefly, serial coronal sections were permeabilized with a 1:1 ratio of cold (−20 °C) acetone and methanol for 10 min, then stained with 0.1% Cresyl Violet containing 0.3% acetic acid. Brain sections were then dehydrated with 100% ethanol and mounted with Permount (Fisher).

**Luxol Fast Blue staining.** Luxol Fast Blue staining was performed to measure the density of myelinated regions. Brain sections were mounted on slides and placed into a 1:1 ethanol:chloroform solution for 6 h. Sections were then incubated in a 0.1% Luxol Fast Blue solution dissolved in 95% ethanol, solvent blue 38 (Millipore Sigma), and 0.5% acetic acid, in a 56 °C incubator overnight. The following day, sections were rinsed of excess stain with 95% ethanol, and differentiated in a 0.1% lithium carbonate solution for 1 min. Brain sections were then dehydrated with 100% ethanol followed by xylene for 5 min. Permount mounting medium was used for microscopy.

**Immunofluorescence.** All brain sections were stored in antifreeze solution (30% ethylene glycol, 30% sucrose in 30% pH 7.4 PBS) at −20 °C prior to immunolabeling. By using the free-floating method, sections were washed with TBS-T (0.1% Tween-20 buffer), and primary antibodies were incubated in blocking solution (0.1% Triton X-100 in SuperBlock TBS buffer, Thermo) at 4 °C overnight. The following day, sections were washed with TBS-T and incubated with secondary fluorescent antibodies (Jackson ImmunoResearch) for 2 h at room temperature. Sections were then washed with TBS-T and mounted onto coverslips using Mowiol (2.5% PVA/DABCO) for imaging.

**Imaging.** Nissl and LFB Images were acquired on a Zeiss AxioScan Z1 slide scanner with a 5× objective. Immunofluorescence images were acquired on an LSM 880 airy confocal system with 20X and 40X objectives using a tile scanning configuration.

**Analysis.** For Nissl analysis, areas were quantified using ZEN software (blue edition). For LFB analysis, ROIs were quantified using Image J software (inverted image). Stereological quantifications of corpus callosum (genu and body) were performed for immunolabeling analysis by using ZEN software.

**CUBIC tissue clearing and LSFM.** Tissues were cleared using CUBIC reagents (TCI, T3740, T3741) as previously described[61,144]. Mice were perfused and brains were harvested and postfixed as described above. On day 2, brains were washed with PBS for 2 h at RT and then immersed in 50% CUBIC-L. On day 3, the solution was replaced with 100% CUBIC-L. On day 5, brains were washed with PBS for 3 h, immersed in 50% CUBIC-R, and changed to 100% CUBIC-R on day 6. On day 8, brains were transferred to CUBIC mounting solution (TCI, RI 1.520, M3294). Images were acquired on a Light Sheet Fluorescence Microscopy (LSFM) at Zeiss Microscopy (Thornwood, NY).

**Transmission electron microscopy sample preparation and processing.** Brain tissue samples were collected from the genu region of the corpus callosum of mice. Specifically, samples were sourced from 3-month-old mice for the Nestin and Olig2 groups, and from 7-month-old mice for the clemastine treatment group. The mice were perfused with a 4% PFA solution. Following perfusion, the tissue samples were

fixed for 1 h at room temperature using a solution comprising 2.5% glutaraldehyde and 2% paraformaldehyde in 0.1 M sodium cacodylate buffer then incubated overnight at 4 °C. The samples underwent postfixation with 1% osmium tetroxide, followed by 2% uranyl acetate. Dehydration was performed using a graded series of ethanol, transitioning from 30% to 100%. The samples were embedded in LX112 resin (LADD Research Industries, Burlington VT). Ultrathin sections were prepared using a Leica Ultracut UC7, stained sequentially with uranyl acetate and lead citrate, and subsequently visualized on a JEOL 1400 Plus transmission electron microscope operated at 120 kV (Albert Einstein College of Medicine Analytical Imaging Facility; SIG # 1S10OD016214-01A1). Digital TEM images were originally acquired as.dm4 files and then converted to 8-bit TIFF files using ImageJ software for compatibility and further processing. The images were then analyzed using hand tracing and photoshop measurement tools. For each mouse, 100 axons were analyzed, and processing was completed while blinded to mouse identity. The Olig2 group was processed with a resolution of 0.00539 um per pixel, while both Nestin and clemastine groups were processed at a resolution of 0.00258 um per pixel. Axons were classified in groups based on their axonal diameter (grouped in microns as 0.1 to 0.29, to 0.4, to 0.51, to 0.66, to 1.7, >1.7).

**Fluorescent In situ hybridization (FISH).** FISH was performed according to the procedure outlined in the Stellaris®FISH kit (Biosearch Technologies, Inc., Petaluma, CA)[145]. Custom Stellaris®FISH Probes were designed against the rat/mouse *Anks1b* gene using the Stellaris®FISH Probe Designer, targeting the 3' region of the *Anks1b* gene to create a specific probe for the larger ankyrin-repeat containing splice variant (AIDA-1B), and targeting the 5' region to create probes for both smaller AIDA-1D and larger AIDA-1B variants. Brain sections were hybridized with FAM-AIDA-1B and CAL Fluor Red 590-AIDA-1D probes. Fixation, pre-hybridization, and hybridization were performed following the manufacturer's protocol.

**Cell culture.** Oligodendrocytes were grown on 18 mm round glass coverslips. Cells were fixed and permeabilized with 70% ethanol for 1 h at 4 °C. Coverslips were washed with Wash buffer A (Stellaris, Biosearch Technologies) for 5 min at room temperature. Cells were then incubated with hybridization buffer containing *Anks1b* probes for 6 hr at 37 °C. Then, coverslips were washed with Wash Buffer A and incubated in the dark at 37 °C for 30 min. After Wash Buffer A was aspirated, cells were stained with DAPI, incubated in the dark at 37 °C for 30 min, and mounted on slides for imaging.

**Fixed tissue.** Sagittal brain sections (5 μm thick) were mounted on slides and fixed with cold electron microscopy grade PFA in PBS for 15 min at RT. They were washed with PBS for 5 min, dipped in nuclease-free water, and dipped in TEA buffer (13.3 mL Triethanolamine, up to 100 mL Nuclease-free water, pH 8.0). Slides were immersed in TEA with acetic anhydride for 10 min with stirring, immersed in in 2x SSC for 3 min, washed sequentially in 70%/95%/100% ethanol for 3 min each, fixed with chloroform for 3 min, washed in 100% and 95% of ethanol for 3 min each, then dried for 90 min at RT. Hybridization Buffer containing *Anks1b* probes was dispensed for 12 h at 37 °C. Slides were immersed in Wash Buffer A and incubated in the dark at 37 °C for 30 min. After Wash Buffer A was aspirated, slides were stained with DAPI and incubated in the dark at 37 °C for 30 min. Slides were immersed in Wash Buffer B for 3 min and immersed sequentially in 50%/85%/100% ethanol for 3 min each. After air drying for 10 min, Prolong Gold Antifade Mounting solution was added and sections were coverslipped for imaging.

**Neuron and oligodendrocyte cultures.** Primary cortical and hippocampal cultures were prepared from Sprague-Dawley rat embryos at E18 as previously described[146]. For mixed cultures containing neurons

and oligodendrocytes, primary cells plated in standard media were switched either to Neurobasal + B27 + 2.5% FBS or Neurobasal and MYM media as described[147]. For enriched oligodendrocyte cultures, brain tissues from mouse (postnatal days 1–5) or rat (Embryonic E20-E22; Postnatal P1-P5) pups were obtained. After decapitation, brains were extracted from skulls, and the cortex and hippocampi were dissected in aseptic conditions. Cerebral cortices were stripped of dura matter in Leibovitz's L-15 Medium supplemented with glucose 6 g/mL (dissection media) and incubated in 1X Trypsin (Gibco; 15–25 min at 37 °C). Tissues were washed of trypsin using dissection media and finally incubated in OPC base media composed of DMEM F12 with glucose 6 g/L, glutamax 2 mM penicillin and streptomycin 50 mg/mL, Neomycin 100 mg/mL, and 1X B27 (Gibco). Cells were then triturated by mechanical means which typically involved passing tissues sequentially through 1 ml, 200 ul and 20 ul tips or using a fire-polished glass pipette. For experiments with mouse OPCs, cortices from each pup were processed separately and incubated in media on ice while PCR genotyping experiments were performed. For rat cells, all cortices were combined and processed together. Following dissociation and genotyping (if necessary), cells were plated and grown on wells/coverslips coated with poly-L-lysine (Sigma-Aldrich, St. Louis, MO, United States) in OPC media supplemented with PDGF-AA 30 mg/mL and FGF 10 mg/mL (growth media). Cells were fed by replacing 1/3 of media typically every 3–4 days. For maturation, which typically occurred 6–10 days following OPC expansion and once sufficient growth was observed, the growth media was replaced by OPC media supplemented with T3 0.75 ng/ml and followed for 5–7 days. Where appropriate, cells were infected with lentivirally delivered shRNAs described previously[7].

**Western blot.** For western blot, the cells were lysed with LDS (Lithium dodecyl sulfate) buffer containing 150 mM tris HCl, 2% LDS, 10% glycerol, 0.51 mM EDTA, 0.22 mM Orange G at pH 8.5 and disaggregated using a syringe with a 28 G ½ needle. The samples were run on precast 4–12% Criterion gels (Bio-Rad Laboratories, Inc., Hercules, CA, United States). After electrophoresis, the gels were transferred onto nitrocellulose membranes for 1 h. The membranes were blocked with 5% dry milk diluted in Tris-buffered saline (TBS) buffer with 0.1% tween 20. Membranes were incubated for 1 h at room temperature with primary antibodies described in Supplementary Data 3. Proteins were visualized with the incubation of fluorescent secondary antibodies and imaged using the Odyssey CLX imaging system (LI-COR Biosciences, Nebraska).

**Oligodendrocyte maturation assay with EdU-labeling.** Mice at different developmental stages were injected IP with Click-iTTM EdU (50 mg/kg, Invitrogen) five times a day (2 h intervals) and sacrificed 4 weeks later as described[148]. For the 3-week-old time point, EdU was injected in pregnant females at E18.5, and pups were sacrificed at P21. In adult (2 and 6 months old), and aged (12 months old) mice, EdU was injected at 2, 6, and 12 months old, and mice were sacrificed 4 weeks later at 3, 7, and 13 months old, respectively. Coronal sections (40 μm thickness) were prepared and processed for immunofluorescence. Images were acquired on a Zeiss LSM 880 airy confocal system with 20× and 40× objectives using a tile scanning configuration. Stereological quantification of EdU+ cells (newborn cells that had migrated into the corpus callosum), EdU+/PDGF1a+ cells (OPCs and immature oligodendrocytes), and EdU+/CC1+ cells (mature oligodendrocytes) were quantified within the corpus callosum area using Zen software.

**Demyelination and remyelination assays.** Mice at 6–8 weeks of age were anesthetized with 5% isoflurane and maintained at 1–2% isoflurane on a stereotaxic apparatus (Stoelting). Mice were given flunixin to manage pain and moisturizing eye ointment was added at the beginning and end of surgery. LPC (Lysophosphatidylcholine, 62963,

Sigma, 1% LPC in 0.9% NaCl) was stereotactically injected into the corpus callosum at 1 site (2 μL per site) at the following coordinates (AP: +1, ML: +1.5, DV: −1.8 mm). On the following day, EdU (50 mg/kg) was injected via IP route 3 times (at 2 h intervals) for 1 day to track the population of newborn cells in the corpus callosum during the de-, and re-myelination phases[116,149]. Mice were sacrificed at 7 days post injection (D, demyelination), 10 dpi (RI, remyelination initiation), and 14 dpi (R, remyelination). Focal demyelinating region was validated with LFB staining and MBP staining. Images were acquired on a Zeiss LSM 880 airy confocal system with 20X and 40X objectives using a tile scanning configuration. Stereological quantification of EdU+ cells (newborn cells migrated into the demyelinated region of the corpus callosum) was quantified using Zen software.

**FRET live imaging of Rac1 activation.** OPCs, obtained as described above, were plated at a cell density of $2 \times 10^5$ on 25 mm glass coverslips. For live-cell imaging, artificial cerebral spinal fluid (ACSF) containing (in mM) 124 NaCl, 26 NaHCO₃, 10 glucose, 2.5 KCl, 1 NaH₂PO₄, 2.5 CaCl₂, and 1.3 MgSO₄ was used. Cells were imaged for a total of 16 minutes, with Rac1 stimulator (CN04 at 40% recommended, Cytoskeleton, inc; IGF-1 at 1000x) added after 4 min. Image acquisitions were performed through a 40× magnification objective lens (UIS 40× 1.30 NA; Olympus) using a custom microscope[37]. This microscope was capable of simultaneous acquisition of FRET and donor mCerulean emissions through two PrimeBSI-Express cameras (Photometrics, Tucson, AZ, USA) that are mounted via a 4-way optical beam splitter (Cairn Research, Kent, UK) and containing a T505LPXR mirror, ET480/40M for mCerulean emission, and ET535/30M for mVenus-FRET emission (Chroma Technology Corp, Bellows Falls, VT, USA). The relative intensities between the two channels were balanced by the inclusion of a neutral density filter (ND0.2 in FRET channel) to ensure that the range of brightness in both mCerulean and FRET channels were similar, to maximize the signal to noise ratio. Cells were illuminated with a 100 W Hg arc lamp through a neutral density filter to attenuate the light as needed and then through an ET436/20X (Chroma Technology) bandpass filter for mCerulean excitation. The main fluorescence turret of the microscope contained a custom T-cfp/yfp/cy5pc-WF dichroic mirror (Chroma Technology). The mCherry fluorescence (a marker of rat cells expressing AIDA-1 shRNAs) was acquired through a PrimeBSI camera (Photometrics) also attached on the third port of the optical beamsplitter (Cairn Research) via the T560LPXRXT mirror (Chroma Technology) and FF628/32 emission filter (Semrock). MetaMorph software ver. 7.10.5 (Molecular Devices) was used to control the microscope, motion control devices, and image acquisition. Metamorph and MatLab (ver 2011a; Mathworks, Natick, MA, USA) software were used to perform image processing and data analyses, as previously described[150–152]. Ratiometric image processing included flatfield correction, background subtraction, image registration, and ratio calculations[153]. In brief, flatfield correction involved the acquisition of cell-free fields of view with the same exposure and field illumination conditions as the foreground image sets (shading images). The foreground images were then divided by the shading images to obtain flatfield-corrected images. A small region of interest in the background (cell-free) area was selected in the flatfield-corrected foreground image sets, and the mean gray value from such a region was subtracted from the whole field of view, calculated, and processed at each time point to obtain the background-subtracted image sets. The background-subtracted image sets were then subjected to an affine transformation based on a priori calibration to account for misalignments between the three cameras used for the simultaneous imaging of the FRET and mCerulean channels, plus the mCherry channel. After the transformation, a linear X−Y registration was performed on the resulting image sets, before ratio calculations, in which the FRET image set was divided by the mCerulean channel

image set. For the imaging of biosensors, we adjusted the intensity of excitation light and the camera acquisition time duration by targeting to fill approximately 50% of the total digitization range of the sCMOS device circuitry, to maximize the dynamic range, using excitation light intensities of 0.4–1.0 mW at the specimen plane.

**Animal behavioral assays.** All experiments were approved by and complied with the Albert Einstein College of Medicine IACUC. All mice were housed and handled at Einstein and behavioral phenotyping was performed as 7 independent cohorts in the Animal Behavior Core under the supervision of Dr. Maria Gulinello. All behavioral testing was performed as previously reported[3]. In the Behavioral Spectrometer, mice were recorded for 9 min (3 min, 3×) in an open field with a center area of 18.0 × 18.0 cm; validity and specificity of the system in identifying ASD-related behaviors has been previously described[154]. Elevated plus maze (5 min), forced swim test for 9 min (3 min training, 4 min test in 25 °C water bath), and three-chamber test (5 min with ovariectomized C57BL/6J mouse, Jax) were performed using standard procedures[155]. Social preference was calculated using the following formula: *social preference = (social sniffing time)/(social sniffing time + object sniffing time) x 100%*. Acoustic startle reflex and prepulse inhibition of the startle response were assayed[156] in a single session for each mouse using randomized, interleaved trials (5 each) for acoustic startle response (115 dB) and prepulse inhibition (PPI). Short-delay PPI trials were conducted with an 81-dB prepulse 40 ms before the 115-dB target stimulus, while long-delay PPI was tested with a 200-ms inter-stimulus interval. Balance beam assay for motor coordination was performed as described[157]. Mice were started at the center of a wood-circular beam (Diameters: 1.2 cm, 1.5 cm, 1.8 cm) after pre-training. The novel object placement test was performed[158] using a 5-min training phase, 4-min testing phase, and short (40 min) retention intervals. During the testing phase, preference for the new placement was calculated using the following formula: *preference for new = (time exploring new placement)/(time exploring new placement + time exploring old placement) x 100%*. All behavioral testing in adult mice aged 3–4 months was performed with experimenter blind to genotype. Feeding, mixed-genotype group housing, light-dark cycles, and time of testing were controlled across all cohorts[159]. Testing chambers were sanitized with 70% ethanol between trials. Mouse movement was recorded by a video tracking software (Viewer) and analyzed by off-line video tracking software (EthoVision XT 14). Sample sizes were estimated in JMP (version 16 and 17, SAS) to be of sufficient power to show effects independently in either sex if main effect of genotype-sex interaction (a = 0.05) were detected in 2-way ANOVA. For all behavioral tests, post hoc *t* test (Tukey) was performed for tests in which genotype was a significant main effect in 2-way ANOVA. For the three-chamber test, Likelihood Ratio effect tests were additionally performed for passing (>50% social preference) or failing (<50% social preference). Post hoc contingency testing included chi-squared analyses and Fisher's exact test. Since no main effect of sex or genotype-sex interaction was observed in any behavioral test, males and females of each genotype were combined for all post hoc testing[159]. Power analysis and least significant number (LSN) were calculated in JMP for post hoc *t* tests on significant effects from 2-way ANOVA analyses. 2-way ANOVAs were performed to detect main effects and interactions of genotype, and sex.

**Clemastine rescue experiments.** For treatment with clemastine fumarate (Cat #:1453; Tocris), the compound was solubilized at 1 mg/ml in a PBS solution containing 10% DMSO and filter sterilized (0.22 um). Clemastine was administered IP at a dose of 10 mg/kg (~270 uL per animal) daily at noon for 2 weeks, alternating the location of injection each day (left or right peritoneum), as has been previously described[17,72,160]. Controls were injected with the same

volume of 10% DMSO/PBS (vehicle). All injections were performed blind to mouse genotype. All injected animals were 7–8 months of age and group-housed according to the experimental group. After the injection period, animals were allowed to recover for 7 days prior to behavioral testing. Social preference and behavioral spectrometer analyses were performed as described above. All mice had been previously tested in these assays at 3–4 months of age.

## Reporting summary
Further information on research design is available in the Nature Portfolio Reporting Summary linked to this article.

## Data availability
All raw data and the datasets generated during and/or analyzed during the current study are available from the corresponding author on request. Source data are provided as a Source Data file. Source data are provided with this paper.

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

## Acknowledgements
Supported by NIH R01NS118820, NIH R56MH115201, NIH 1UL1TR002556-01 and Simons AR-PI-0000251 to B.A.J., NIH T32HD098067 to A.M., NIH T32GM007288 to A.U.C., R35GM136226 to L.H., R01NS123374 and R01NS082432 to M.L.L. and SIG #1S10OD016214 and P30CA01330 to the Analytical Imaging Facility at Albert Einstein College of Medicine. Significant support for this work came from the Rose F. Kennedy Intellectual and Developmental Disabilities Research Center (IDDRC), which is funded through a center grant from the Eunice Kennedy Shriver National Institute of Child Health & Human Development (NICHD U54HD090260). We thank Xheni Nishku, Leslie Cummins, and Joseph Churaman in the Analytical Imaging Facility at the Albert Einstein College of Medicine. We thank the participating families for their time and contributions to this work.

## Author contributions
C.H.C. developed the idea that AIDA-1 regulates oligodendrocyte function. C.H.C. performed all histology and immunofluorescence to measure myelin and OPC development in tissue, as well as behavioral analyses of Olig2 Het mice, which were created and maintained by C.H.C. and I.V.D. I.V.D. assisted in histology and immunofluorescence assays and developed the idea that AIDA-1 regulates Rac1. I.V.D. performed all proteomic analyses and Rac1 activity measurements together with L.H. I.V.D. and D.C.M. performed all EM experiments and quantitation in diverse Anks1b mouse models. D.C.M., J.V. and B.A.J. performed clemastine rescue assays. J.R.D., A.S.M. and B.A.J. performed in vitro assays in primary oligodendrocyte cultures. A.U.C. and J.O.T. performed behavioral analyses on CaMKII and L7 Anks1b mouse models. R.F., M.H.C., J.V., C.A.B. and M.L.L. performed and analyzed MRI and DTI experiments on ANDS patients and mouse models. S.M. was involved in the recruitment of patients and MRI analyses. C.H.C., I.V.D. and B.A.J. wrote the manuscript.

## Competing interests
The authors declare no competing interests.
