## [Peer Review File · Nature Communications]

ANKS1B encoded AIDA-1 regulates social behaviors by controlling oligodendrocyte functionREVIEWER COMMENTS

Reviewer #1 (Remarks to the Author):

In this study, Cho et al. reported novel and interesting findings that expression of AIDA-1, encoded by ANKS1B gene in oligodendroglia, resulted in abnormal oligodendrocyte phenotypes, including impaired OPC maturation, reduced oligodendrocyte abundance, and myelination. These cellular phenotypes further led to structural abnormalities in the corpus callosum and impaired social behavior of the mice. It is intriguing that only heterozygous deletion (Olig2-Het) and Olig2-knockout of ANKS1B gene in mainly oligodendroglia, but not in forebrain excitatory neurons (CamK2a-Cre) or cerebellar Purkinje neurons (L7-Cre), cause deficits in social behavior. Furthermore, the authors showed that clemastine fumarate treatment rescued impaired social behavior in adult mice lacking ANKS1B in oligodendrocytes. These findings suggest a significant contribution of oligodendroglia dysfunction to autism-associated behavioral deficits. There are some concerns that need to be considered.

1. The main weakness of this work is the lack of mechanistic studies to explore how ANKS1B deficiency could cause the oligodendroglial phenotypes. Some interesting ideas have been discussed, such as working through Rho family GTPases, neuron-OPC synaptic junctions and signal transduction etc. However, there were no data directly showing changes in Rho family GTPases or whether manipulating those GTPases would help with the oligodendroglial phenotypes in the heterozygous deletion or knockout mice. Alternatively, it would also be helpful to have some data that indicate the involvement of AIDA-1 in neuron-OPC synapses, for example, whether AIDA-1+ puncta on oligodendroglia are synapses, whether AIDA-1 show activity-dependent translocation in OPCs etc.
2. More thorough characterizations of myelin phenotypes would be needed. For myelin deficits, typically electron microscopy analysis of myelin sheath and accompanied g-ratio analysis should be performed.
3. Did Anks1b haploinsufficiency in Nestin-Het and Olig2-Het mice, and deletion in Olig2-KO mice alter astroglial differentiation, considering that nestin and Olig2 are expressed by neural progenitor cells that can also give rise to astroglia, in addition to neurons? Using additional oligodendroglia-specific cre lines, such as PDGFRalpha-Cre would be helpful to support the conclusion that AIDA-1 has cell-autonomous effects in oligodendroglial maturation and function.
4. Data that demonstrate the decreased expression or knock out of AIDA-1 in oligodendroglia in Nestin-Het, Olig2-Het, and Olig2-KO mice should be shown.
5. In Fig. 5, it is admirable that human OPCs were used. However, those human induced OPCs (iOPCs) were not clearly defined. Were these human iOPCs differentiated from human induced pluripotent stem cells or human iOPCs induced by reprogramming methods with transcription factors? In addition, there were no data showing the identity of the human iOPCs. The authors described in the figure legend that "Olig2 antibodies used do not recognize human Olig2 in iOPCs". In fact, there are olig2 antibodies that can recognize both mouse and human Olig2, for example, Uchida et al. *Sci Transl Med* 2012
6. There were no histological data showing oligodendroglia and myelin following clemastine rescue? Staining of myelin markers should be shown. Electron microscopy analysis should be done as well.

Reviewer #2 (Remarks to the Author):

This paper shows that oligodendrocyte dysfunction caused by mutation of Anks1b, one of the risk genes for neurodevelopmental disorders, is associated with behavioral abnormalities such as decreased social behavior. The authors generated mice heterozygous for Anks1b by Nestin-Cre and showed that they showed structural abnormalities in the corpus callosum and decreased

myelin formation. Furthermore, using mice heterozygous for Anks1b by Olig2-Cre, the authors found that not only myelin dysplasia but also behavioral abnormalities such as decreased social behavior are oligodendrocyte autonomous. It was also shown that administration of clemastine fumarate, which promotes myelin formation, ameliorated the decline in social behavior of mutant mice. This is an interesting study showing that abnormalities in oligodendrocyte function and myelin formation may have important effects on neurodevelopmental disorders. However, there are some problems to be solved as follows.

1) MRI data show that the volume and fractional anisotropy of the corpus callosum are reduced by Anks1b heterozygous deficiency. Is it similarly reduced in areas other than the corpus callosum? Is there region specificity?

2) The authors have shown that Anks1b heterozygous deficiency causes increased proliferation and impaired differentiation at 3 weeks to 12 months of age. Since oligodendrocyte proliferation and differentiation are active in the first few weeks of life, the number of oligodendrocyte precursor cells and oligodendrocytes and the degree of proliferation and differentiation at the earlier developmental stage (within 3 weeks of age) should be indicated.

3) It is necessary to show the molecular mechanism of how Anks1b regulates proliferation, differentiation and myelin formation. The authors have described potential interesting molecular functions in the discussion, but experimental data should be presented.

4) As stated by the authors, Olig2-Cre has been reported to be expressed in some neurons, which makes it difficult to interpret the results of this study. Reporter mice should be used to show the proportion of Olig2-Cre expressed in neurons and astrocytes in brain regions such as the corpus callosum, cerebral cortex, and cerebellum. Is there any reason not to use other oligodendrocyte lineage-specific Cre lines such as NG2-Cre and Cnp-Cre?

5) The effects of heterozygous and homozygous Anks1b using Olig2-Cre on oligodendrocytes should be compared. In Fig. 6d, the number of oligodendrocytes was decreased to the same extent in heterozygous and homozygous mice, but what is the reason for the lack of gene dosage effect?

6) Since there is no change in the expression level of interneuron markers and the number of Purkinje cells in Olig2-Cre/Anks1b heterozygous-deficient mice (Supplementary Fig. 7b, d), the authors consider that these cells have little effect on behavior. However, this alone is not enough. Given that Anks1b functions at synapses, the synaptic function of these cells may be impaired and this should be examined in detail.

7) The authors show that Anks1b-deficient mice using L7-Cre and CamK2-Cre do not show behavioral abnormalities (Fig. 7d). Anks1b is thought to regulate NMDA receptor function, but are synaptic dysfunction seen in CamK2-Cre/Anks1b-deficient mice? In addition, Nestin-Cre/Anks1b homozygous knockout mice die shortly after birth, suggesting that Anks1b also plays an important role in neurogenesis. In this case, analysis using postnatally expressed L7-Cre and CamK2-Cre is likely to show no phenotype, which is insufficient to analyze the effects of Anks1b deficiency.

8) The result that clemastine improves sociality is very interesting. It should be shown whether there is also an improvement in the number of oligodendrocytes and myelin formation. Does administration of clemastine also improve the phenotype of Nestin-Cre/Anks1b heterozygous mice?

Reviewer #3 (Remarks to the Author):

This paper by Cho C-H et al reports an unexpected role for ANKS1B in oligodendrocyte (OLG) function and a potential role for OLGs in pathophysiology of ANKS1B syndrome using the mouse model.

The purpose of this research is very interesting and novel, and most of the data are good to

explain their hypothesis. However, this reviewer has several critical points that might improve this paper as follows.

Major

1, Although many immunohistochemical evidences are performed to show less OLG maturation and the dysfunction, electrophysiological experiment should be added to prove less axonal conduction velocity in the corpus callosum of the model. In addition, the treatment of clemastine will result in the normalization of the conduction velocity. Thus, further electrophysiological data will strengthen this paper.

2, Although Nestin-Het mice are used as the model of human ANKS1B syndrome, unfortunately, there is no direct evidence of the relationship between AIDA-1 protein expression level and OLG dysfunction in this paper. As the symptom of neurodevelopmental disease shows spectrum, it is very important to show the relationship between AIDA-1 protein expression and behavior in Olig2-Cre driven Anks1b conditional mutant mouse.

Minor

1, In Fig2: It is happy for us to have western data of AIDA-1 protein in abnormal myelin sheath and reduced OLGs density in Anks1b haploinsufficiency mouse model.

2, In Fig4: Do you have some information of the relationship between AIDA-1 protein and maturation and/or migration?

3, In Fig7: AIDA-1B protein level should be investigated in the Olig2-Cre, L7-Cre, and CaMKII α -Cre mice to know whether the expression levels are related to the behavioral tests, especially social preference.

4, Please check the protein level of AIDA-1B in clemastine treatment.

We want to thank the reviewers for their valuable comments and apologize for the delay in our resubmission. It took
some time to address the important concerns raised, which involved incorporating new techniques into our laboratory.
Major improvements to the manuscript include a detailed mechanistic investigation and a multi-faceted verification of
myelination deficits in *Anks1b* models. Using advanced G-protein biosensors, we investigated causative mechanisms
linking AIDA-1 to oligodendroglial phenotypes. We find that AIDA-1 regulates Rac1 activity and that stimulating Rac1
function can improve the deficits in oligodendrocyte function present in *Anks1b*-deficient mice. We also employed
electron microscopy (EM) to confirm the presence of myelination deficits in mouse models and show that clemastine
was effective in rescuing both myelination and social deficits in these models.

Below, we answer all concerns expressed by **Reviewer #1**, **Reviewer #2**, and **Reviewer #3**.

**Reviewer #1 (Remarks to the Author):**

In this study, Cho et al. reported novel and interesting findings that expression of AIDA-1, encoded by ANKS1B gene in
oligodendroglia, resulted in abnormal oligodendrocyte phenotypes, including impaired OPC maturation, reduced
oligodendrocyte abundance, and myelination. These cellular phenotypes further led to structural abnormalities in the
corpus callosum and impaired social behavior of the mice. It is intriguing that only heterozygous deletion (Olig2-Het) and
Olig2-knockout of ANKS1B gene in mainly oligodendroglia, but not in forebrain excitatory neurons (CamK2a-Cre) or
cerebellar Purkinje neurons (L7-Cre), cause deficits in social behavior. Furthermore, the authors showed that clemastine
fumarate treatment rescued impaired social behavior in adult mice lacking ANKS1B in oligodendrocytes. These findings
suggest a significant contribution of oligodendroglia dysfunction to autism-associated behavioral deficits. There are
some concerns that need to be considered.

**1. The main weakness of this work is the lack of mechanistic studies to explore how ANKS1B deficiency could cause the**
**oligodendroglial phenotypes. Some interesting ideas have been discussed, such as working through Rho family GTPases,**
**neuron-OPC synaptic junctions and signal transduction etc. However, there were no data directly showing changes in**
**Rho family GTPases or whether manipulating those GTPases would help with the oligodendroglial phenotypes in the**
**heterozygous deletion or knockout mice. Alternatively, it would also be helpful to have some data that indicate the**
**involvement of AIDA-1 in neuron-OPC synapses, for example, whether AIDA-1+ puncta on oligodendroglia are synapses,**
**whether AIDA-1 show activity-dependent translocation in OPCs etc.**

This concern represented the focus of our revisions. To search for potential mechanisms linking AIDA-1 to
oligodendroglial phenotypes, we conducted a comparative analysis by cross-referencing our previously
published proteomic studies identifying the AIDA-1 interactome (Carbonell AU 2019) and synaptic proteome of
*Anks1b*-deficient mice (Carbonell AU 2023). This examination identified a potential link to Rho family GTPases.
Using FRET-based G-protein biosensors, we found that AIDA-1 regulates Rac1 activity, as shown in **NEW Figure 6**
**(h, i) and NEW Supplementary Figure 10 (a, b)**. Moreover, we find that stimulating Rac1 activity can effectively
restore deficits in maturation in oligodendrocytes derived from *Anks1b* models, as shown in **NEW Figure 6 (j, k)**
**and NEW Supplementary Figure 10 (f)**. This discovery is significant because previous research has established
the crucial role of Rho family GTPases, and particularly Rac1, in the maturation of OPCs and myelination, which
are the primary deficiencies observed in *Anks1b*-deficient mice.

**2. More thorough characterizations of myelin phenotypes would be needed. For myelin deficits, typically electron**
**microscopy analysis of myelin sheath and accompanied g-ratio analysis should be performed.**

We conducted the suggested experiments and are happy to report that the results validate our main
conclusions. In **NEW Figure 2 (c, d)**, electron microscopy and g-ratio analyses of callosal axons reveal clear
myelination deficits in Nestin-Het mice, and in **NEW Figure 5 (g, h)**, similar analyses confirm myelination deficits
in Olig2-Het mice.

**3. Did *Anks1b* haploinsufficiency in Nestin-Het and Olig2-Het mice, and deletion in Olig2-KO mice alter astroglial**
**differentiation, considering that nestin and Olig2 are expressed by neural progenitor cells that can also give rise to**
**astroglia, in addition to neurons?**

Western blots in **NEW Figure 4 (d)** confirm that AIDA-1 is not expressed in astrocytes. This result validates
findings from multiple single cell transcriptomic databases like Tabula Muris, The Allen Institute Cell Atlas, and
the Broad Institute (**Supplementary Figure 4**), which show that *Anks1b* transcripts are not expressed in

astrocytes. Therefore, any potential deficits would likely have indirect causes, although casual observations did
not reveal any obvious astrocytic deficits using GFAP staining in cultures and slices.

**Using additional oligodendroglia-specific cre lines, such as PDGFRalpha-Cre would be helpful to support the conclusion**
**that AIDA-1 has cell-autonomous effects in oligodendroglial maturation and function.**

We understand the suggestion, but limitations associated with all promoters used in oligodendrocyte research
weakened the arguments needed to overcome the temporal and financial constraints associated with creating
another mouse model. It is important to note that increasingly detailed single-cell transcriptomic experiments
and evidence of transient promoter expression in diverse cell lines reveal that even promoters previously
confirmed as “specific” may not exhibit absolute specificity (Tognatta R, 2016). For example, the Allen Brain Cell
Atlas, DropViz, and Tabula Muris databases all show that PDGFRA is expressed in diverse interneurons of the
CNS as well as in different organs throughout the body, which would be problematic. Other oligodendrocyte
promoters such as NG2 can be found in various cell types, including astrocytes, pericytes, and diverse cells
throughout the body, and NG2-positive cells have been found to adopt neuronal markers both *in vitro* and *in*
*vivo*. CNPase has also been implicated in microglial function, and its expression has been detected in different
organs, including astrocytes in the brain. CNP-GFP has also been found in Nestin+ but NG2- cells in culture,
which suggests a neuronal expression. It is noteworthy that among commonly used oligodendrocyte promoters,
Olig2 shows the highest specificity for oligodendrocyte lineage cells based on single-cell transcriptomic
databases. However, we agree that this is a caveat to consider in our work and have expanded our discussion to
address these complexities and potential limitations in our study.

Therefore, to address these concerns we have conducted a series of experiments to test the oligodendrocyte-
autonomous nature of the observed phenotypes and rescues. Here are the key findings:

Specific Reduction of AIDA-1 in oligodendrocytes: In **New Figure 5 (i, j)**, Western blots show that AIDA-1 protein
expression is selectively decreased in oligodendrocytes but not in neurons isolated from Olig2-Het mice. No
expression is observed in astrocytes.

No changes in AIDA-1 levels in total brain lysates: The same Western blots reveal no discernable differences in
AIDA-1 protein expression in total brain lysates between genotypes. Because heterozygosity in a small subset of
cells would not be detectable by Western blot, this result is consistent with specific recombination in OPCs and
oligodendrocytes that represent only 5-10% of all brain cells.

Specific deficits in isolated oligodendrocytes: Imaging studies and Western blots in **New Figure 6 (a, b) and**
**Supplementary Figure 10 (c, d)** show that oligodendrocytes isolated from Olig2-Het mice, as well as rat
oligodendrocytes treated with AIDA-1 specific shRNAs, exhibit deficits in maturation and MBP production.

Clemastine Rescue: Imaging studies and Western blots in **New Figure 8 (a, b), and Supplementary Figure 10 (f)**
shows that clemastine, an antihistamine that improves myelination by enhancing OPC maturation, effectively
rescues the maturation deficits in primary oligodendrocytes isolated from Olig2-Het mice. This is observed in the
absence of neurons.

In summary, these experiments corroborate our assertion that oligodendrocyte-autonomous dysfunctions are
responsible for the myelination deficits observed in Olig2-Het mice. Together with our findings that clemastine
rescues behavioral deficits in Olig2-Het mice, we present strong evidence that deficits in oligodendrocytes and
myelination are the root cause of the observed social deficits.

**4. Data that demonstrate the decreased expression or knock out of AIDA-1 in oligodendroglia in Nestin-Het, Olig2-Het,**
**and Olig2-KO mice should be shown.**

As discussed above, we have now conducted these important control experiments. Western blots in **NEW Figure**
**5 (i, j)** reveal that AIDA-1 expression is specifically reduced in oligodendrocytes isolated from Olig2-Cre mice.
These results confirm haploinsufficiency in this model. In addition, we find that reduced AIDA-1 expression is
unique to oligodendrocytes and is not observed in neurons.

**5. In Fig. 5, it is admirable that human OPCs were used. However, those human induced OPCs (iOPCs) were not clearly**
**defined. Were these human iOPCs differentiated from human induced pluripotent stem cells or human iOPCs induced by**
**reprogramming methods with transcription factors? In addition, there were no data showing the identity of the human**

iOPCs. The authors described in the figure legend that “Olig2 antibodies used do not recognize human Olig2 in iOPCs”. In
fact, there are olig2 antibodies that can recognize both mouse and human Olig2, for example, Uchida et al. *Sci Transl*
*Med* 2012.

We have removed the human OPCs from our manuscript as their inclusion did not contribute significantly to our
findings. Additionally, it is worth noting that the company (Tempo Bioscience) was unable or unwilling to
disclose the methodology used in the generation of these human iOPCs.

6. There were no histological data showing oligodendroglia and myelin following clemastine rescue? Staining of myelin
markers should be shown. Electron microscopy analysis should be done as well.

We conducted the more conclusive EM studies proposed and in **NEW Figure 8 (h, i)** show that clemastine
treatment, which rescues social deficits in Olig2-Het mice, also leads to an increase in myelination as measured
by EM and g-ratio analyses of callosal axons.

Reviewer #2 (Remarks to the Author):

This paper shows that oligodendrocyte dysfunction caused by mutation of *Anks1b*, one of the risk genes for
neurodevelopmental disorders, is associated with behavioral abnormalities such as decreased social behavior. The
authors generated mice heterozygous for *Anks1b* by Nestin-Cre and showed that they showed structural abnormalities
in the corpus callosum and decreased myelin formation. Furthermore, using mice heterozygous for *Anks1b* by Olig2-Cre,
the authors found that not only myelin dysplasia but also behavioral abnormalities such as decreased social behavior are
oligodendrocyte autonomous. It was also shown that administration of clemastine fumarate, which promotes myelin
formation, ameliorated the decline in social behavior of mutant mice. This is an interesting study showing that
abnormalities in oligodendrocyte function and myelin formation may have important effects on neurodevelopmental
disorders. However, there are some problems to be solved as follows.

1) MRI data show that the volume and fractional anisotropy of the corpus callosum are reduced by *Anks1b* heterozygous
deficiency. Is it similarly reduced in areas other than the corpus callosum? Is there region specificity?

We apologize if this was not clear, but these data had been included in **Supplementary Table 1**. In summary, this
table shows notable deficits in specific brain regions of mice, as revealed by both diffusion tensor imaging (DTI)
and MRI. Furthermore, we now include an extensive DTI analysis of two human patients, comparing their results
to a standardized group of 101 neurotypical controls using a robust analytical pipeline (**NEW Figure 1 (b-d)**).

These rigorous findings highlight significant region-specific abnormalities of white matter microstructure in the
brains of patients. These important human experiments were conducted in collaboration with the renowned
radiologist Michael Lipton, who is widely known for his leadership in the Einstein Soccer study showing the
potential for brain injury from headings in soccer.

2) The authors have shown that *Anks1b* heterozygous deficiency causes increased proliferation and impaired
differentiation at 3 weeks to 12 months of age. Since oligodendrocyte proliferation and differentiation are active in the
first few weeks of life, the number of oligodendrocyte precursor cells and oligodendrocytes and the degree of
proliferation and differentiation at the earlier developmental stage (within 3 weeks of age) should be indicated.

**Figure 3 (b, c)** and **Supplementary Fig 3 (a)** provides precise data on the number of oligodendrocyte lineage cells
148 per cubic millimeter in the corpus callosum for all ages tested. The figures provide specific counts for CC1-
149 positive cells (oligodendrocytes) and PDGFR α -positive cells (OPCs) at these same ages. The 3-week time point
reflects the number of cells that express EdU within the month after pregnant females were injected at
embryonic age of 18.5 days, which covers several critical windows in oligodendrocyte development. Results
show that there was an average decrease of over 1000 CC1+ cells (indicating oligodendrocytes) and an increase
of over 1000 PDGFR α + cells (indicating OPCs) per cubic mm in the Nestin-Het mice.

3) It is necessary to show the molecular mechanism of how *Anks1b* regulates proliferation, differentiation and myelin
formation. The authors have described potential interesting molecular functions in the discussion, but experimental data
should be presented.

We agree this was a weakness in our work and so performed major experiments to identify mechanisms linking
*Anks1b* to oligodendroglial phenotypes. Briefly, we found that AIDA-1 regulates Rac1 activity and that Rac1

stimulation can restore maturation deficits found in oligodendrocytes isolated from *Anks1b* Olig2-Het mice.
Please see our response to point 1 from Reviewer 1 (lines 30-39) and see **New Figure 6**.

4) As stated by the authors, Olig2-Cre has been reported to be expressed in some neurons, which makes it difficult to
interpret the results of this study. Reporter mice should be used to show the proportion of Olig2-Cre expressed in
neurons and astrocytes in brain regions such as the corpus callosum, cerebral cortex, and cerebellum. Is there any
reason not to use other oligodendrocyte lineage-specific Cre lines such as NG2-Cre and Cnp-Cre?

To address this concern, we performed experiments to determine whether results observed were due to
oligodendrocyte-autonomous effects in lieu of generating additional mouse models (Please see our response to
point 3 from Reviewer 1 (lines 59-95)). We also refer the reviewer to **NEW Figure 4 (d)**, which represents a
Western blot confirming that AIDA-1 is not expressed in astrocytes, and **New Figure 5 (i, j)**, which reveals that
AIDA-1 protein expression is selectively decreased in oligodendrocytes but not in neurons isolated from Olig2-
Het mice.

5) The effects of heterozygous and homozygous *Anks1b* using Olig2-Cre on oligodendrocytes should be compared. In
Fig. 6d, the number of oligodendrocytes was decreased to the same extent in heterozygous and homozygous mice, but
what is the reason for the lack of gene dosage effect?

We appreciate the question but currently lack an explanation for the observed lack of gene dosage effect in
experiments counting oligodendrocytes in slices (**Fig. 5 (d)**), while apparent gene dosage effects are evident in
behavioral experiments (**Fig. 7 (b, c, d)**). The focus of this study has been on Nestin and Olig2 heterozygotes. This
was done to investigate haploinsufficiency, as all identified patients are heterozygous for the microdeletions.

6) Since there is no change in the expression level of interneuron markers and the number of Purkinje cells in Olig2-
Cre/*Anks1b* heterozygous-deficient mice (Supplementary Fig. 7b, d), the authors consider that these cells have little
effect on behavior. However, this alone is not enough. Given that *Anks1b* functions at synapses, the synaptic function of
these cells may be impaired and this should be examined in detail.

It is important to note that we neither believe nor do we argue that neuronal deficits are dispensable in the
pathophysiology of ANDS. Patients are ubiquitously heterozygous and we have previously shown that AIDA-1
regulates NMDAR function and synaptic plasticity in the hippocampus (Tindi et al 2015), which would be
expected to influence cognitive and affective behaviors. Indeed, we show that Olig2-Het mice do not
recapitulate all behaviors found in Nestin-Het mice. Our interpretation is that there are cell-specific
contributions to the behavioral deficits found in mouse model of ANDS and here we have focused on social
preference.

However, we recognize that a limitation of our work is that we cannot fully rule out the influence of a small set
of interneurons and have now stated this in the discussion. This subset would be small because Western blot
analyses failed to reveal any changes in AIDA-1 brain expression in total lysate between genotypes. Therefore,
we would first need to identify this subset of neurons to be able to perform electrophysiological tests, which
would be a daunting task. However, it is important to note that even if identified, defining the origins of synaptic
effects may be difficult because changes in myelination could affect synaptic plasticity through input
desynchronization and changes in conduction speeds. Indeed, recent and compelling papers highlight the vital
role of myelination dynamics in memory consolidation, a process that requires synaptic plasticity.

Our study provides robust causal evidence linking myelination deficits to social deficits in Olig2-Het mice, since
clemastine, an anti-histamine that can rescue OPC maturation and myelination, effectively restores myelination
and normalizes social behaviors in Olig2-Het mice. Overall, the strength of our study is that we provide
compelling evidence that both human patients and mice exhibit impaired myelination and provide strong
evidence that deficits in social behaviors present in ANDS originate from oligodendrocyte dysfunction.

7) The authors show that *Anks1b*-deficient mice using L7-Cre and CamK2-Cre do not show behavioral abnormalities (Fig.
7d). *Anks1b* is thought to regulate NMDA receptor function, but are synaptic dysfunction seen in CamK2-Cre/*Anks1b*-
deficient mice? In addition, Nestin-Cre/*Anks1b* homozygous knockout mice die shortly after birth, suggesting that

*Anks1b* also plays an important role in neurogenesis. In this case, analysis using postnatally expressed L7-Cre and
CamK2-Cre is likely to show no phenotype, which is insufficient to analyze the effects of *Anks1b* deficiency.

We previously published that CamK2a-Cre *Anks1b* mice do have a phenotype, which includes impaired NMDAR
signaling and plasticity in the hippocampus (Tindi JO et al 2015). Also, as discussed in our response to point 6, we
do not believe that oligodendrocyte deficits are the only cause of the pathophysiology of ANDS. We agree that
*Anks1b* likely plays a critical role in development because no homozygous ANDS patient has been identified and
Nestin-KO mice are indeed lethal. However, in **Supplementary Figure 3 (b)**, we perform BrdU experiments to
measure neurogenesis in both subventricular zone and subgranular zones but see no difference in cell birth
between Nestin-Het mice and WT.

8) The result that clemastine improves sociality is very interesting. It should be shown whether there is also an
improvement in the number of oligodendrocytes and myelin formation. Does administration of clemastine also improve
the phenotype of Nestin-Cre/*Anks1b* heterozygous mice?

We acknowledge that this was a weakness in our previous submission and have performed EM analyses to
evaluate the outcomes of clemastine treatment. In **NEW Figure 8 (h, i)** we show electron microscopy and g-ratio
analyses of callosal axons in mice that received clemastine treatment. Results show that in addition to rescuing
social deficits in Olig2-Het mice, clemastine also leads to an increase in myelination.

We have not tested the effects of clemastine on Nestin-Het mice but are doing these experiments in a followup
study testing the effects of clemastine of diverse mouse models of ASD.

**Reviewer #3 (Remarks to the Author):**

This paper by Cho C-H et al reports an unexpected role for ANKS1B in oligodendrocyte (OLG) function and a potential
role for OLGs in pathophysiology of ANKS1B syndrome using the mouse model.

The purpose of this research is very interesting and novel, and most of the data are good to explain their hypothesis.
However, this reviewer has several critical points that might improve this paper as follows.

**Major**

1, Although many immunohistochemical evidences are performed to show less OLG maturation and the dysfunction,
electrophysiological experiment should be added to prove less axonal conduction velocity in the corpus callosum of the
model. In addition, the treatment of clemastine will result in the normalization of the conduction velocity. Thus, further
electrophysiological data will strengthen this paper.

We agree that this would be a good functional outcome of our observations. We have instead used behavioral
outcomes as our highest-order readout for the observed myelination defects. Overall, the strength of our study
is that we provide compelling evidence that both human patients and mice exhibit impaired myelination and
provide strong evidence that deficits in social behaviors present in ANDS originate from oligodendrocyte
dysfunctions.

2, Although Nestin-Het mice are used as the model of human ANKS1B syndrome, unfortunately, there is no direct
evidence of the relationship between AIDA-1 protein expression level and OLG dysfunction in this paper. As the
symptom of neurodevelopmental disease shows spectrum, it is very important to show the relationship between AIDA-1
protein expression and behavior in Olig2-Cre driven *Anks1b* conditional mutant mouse.

We completely agree with this assessment and have now conducted these important control experiments.
Western blots In **NEW Figure 5 (i, j)** reveal that AIDA-1 expression is specifically reduced in oligodendrocytes
isolated from Olig2-Cre mice. These results confirm haploinsufficiency in this model. In addition, we find that
reduced AIDA-1 expression is unique to oligodendrocytes and is not observed in neurons, which significantly
bolsters our assertion of the specific targeting of oligodendrocyte lineage cells using Olig2-Cre lines.

**Minor**

1, In Fig2: It is happy for us to have western data of AIDA-1 protein in abnormal myelin sheath and reduced OLGs density
in *Anks1b* haploinsufficiency mouse model.

As described above, please see **New Figure 5 (i,j)**, where Western blots show reduced AIDA-1 expression
specifically in oligodendrocytes isolated from Olig2-Het mice. Also, please see our previous publication
(Carbonell AU 2019, Nature Comms) where we show AIDA-1 haploinsufficiency in the Nestin-Het mice.

**2, In Fig4: Do you have some information of the relationship between AIDA-1 protein and maturation and/or migration?**

Western blots in **NEW Supplementary Figure 10 (c)** show that AIDA-1 levels remain relatively stable (or perhaps
increase slightly) in maturing OPCs/oligodendrocytes in culture.

**3, In Fig7: AIDA-1B protein level should be investigated in the Olig2-Cre, L7-Cre, and CaMKIIa-Cre mice to know whether**
**the expression levels are related to the behavioral tests, especially social preference.**

Please see our previous publication (Tindi JO et al 2015, Journal of Neuroscience) where we show that AIDA-1 is
decreased in the CaMKII α -Cre model, as well as our previous publication (Carbonell AU et al 2019, Nature
communications) where we show that AIDA-1 is decreased in Nestin-Het mice. Also please see our response
your major point 2. **New Figure 5 (i,j)** shows that AIDA-1 is reduced specifically in oligodendrocytes in Olig2-Het
mice, which confirms that social preference deficits are correlated with reduced AIDA-1 in oligodendrocytes.

**4, Please check the protein level of AIDA-1B in clemastine treatment.**

We performed Western blots to test this and in **NEW Supp Fig 10 (f)** show that clemastine has no effects on the
expression of AIDA-1, Rac1, or Olig2.

REVIEWERS' COMMENTS

Reviewer #2 (Remarks to the Author):

I have no additional comments as the content has been greatly improved and the molecular mechanism is also shown.

Reviewer #3 (Remarks to the Author):

This paper by Cho C-H et al. well revised with new technique, and most of my questions and other reviewer's questions are answered with further experiments and additional explanation in the discussion. This reviewer has no more comments and now recommends the acceptance to this high level of journal.